# Individuals with anxiety and depression use atypical decision strategies in an uncertain world

**Zeming Fang[1,2], Meihua Zhao[3], Ting Xu[4], Yuhang Li[5], Hanbo Xie[6], Peng Quan[7], Haiyang Geng[8], Ru-Yuan Zhang[1,2,9]***

[1]Shanghai Mental Health Center, School of Medicine, Shanghai Jiao Tong University, Shanghai, China; [2]School of Psychology, Shanghai Jiao Tong University, Shanghai, China; [3]School of Psychology, South China Normal University, Guangzhou, China; [4]The Center of Psychosomatic Medicine, Sichuan Provincial Center for Mental Health, Sichuan Provincial People's Hospital, University of Electronic Science and Technology of China, Chengdu, China; [5]Centre of Centre for Cognitive and Brain Sciences, Institute of Collaborative Innovation, University of Macau, Macau, China; [6]Department of Psychology, University of Arizona, Tucson, United States; [7]School of Humanities and Management, Guangdong Medical University, Dongguan, China; [8]Tianqiao and Chrissy Chen Institute for Translational Research, Shanghai, China; [9]Shanghai Key Laboratory of Mental Health and Psychological Crisis Intervention, School of Psychology and Cognitive Science, East China Normal University, Shanghai, China

**\*For correspondence:**
ruyuanzhang@sjtu.edu.cn

**Competing interest:** The authors declare that no competing interests exist.

**Abstract** Previous studies on reinforcement learning have identified three prominent phenomena: (1) individuals with anxiety or depression exhibit a reduced learning rate compared to healthy subjects; (2) learning rates may increase or decrease in environments with rapidly changing (i.e. volatile) or stable feedback conditions, a phenomenon termed *learning rate adaptation*; and (3) reduced learning rate adaptation is associated with several psychiatric disorders. In other words, multiple learning rate parameters are needed to account for behavioral differences across participant populations and volatility contexts in this flexible learning rate (FLR) model. Here, we propose an alternative explanation, suggesting that behavioral variation across participant populations and volatile contexts arises from the use of mixed decision strategies. To test this hypothesis, we constructed a mixture-of-strategies (MOS) model and used it to analyze the behaviors of 54 healthy controls and 32 patients with anxiety and depression in volatile reversal learning tasks. Compared to the FLR model, the MOS model can reproduce the three classic phenomena by using a single set of strategy preference parameters without introducing any learning rate differences. In addition, the MOS model can successfully account for several novel behavioral patterns that cannot be explained by the FLR model. Preferences for different strategies also predict individual variations in symptom severity. These findings underscore the importance of considering mixed strategy use in human learning and decision-making and suggest atypical strategy preference as a potential mechanism for learning deficits in psychiatric disorders.

## eLife assessment

This study provides a novel and **valuable** alternative explanation for volatility-induced changes in choice behavior, commonly attributed to learning-rate adaptations. Through rigorous and comprehensive computational modeling of previously published data, the authors provide **convincing**

support for the claim that apparent learning-rate adaptations may instead reflect a mixture of decision strategies. Furthermore, they demonstrate that differential weighting of the optimal decision strategy is predicted by psychopathology common to depression and anxiety. This work should be of interest to a wide range of scientists, including psychologists, neuroscientists, computer scientists, and clinicians.

## Introduction

Intelligent behavior requires the ability to adapt to an ever-changing environment. For example, foraging animals must be able to track the changing abundance or scarcity of food resources in different locations and at different timescales. Motor control demands the ability to control limbs that constantly vary in their dynamics (due to fatigue, injury, growth, etc.). Human competitors in all kinds of games or sports must be able to learn and adapt to their opponents' changing strategies. To understand the mechanisms of these abilities, researchers have examined how (and how well) human agents can learn option values and track the dynamic changes in values in a *volatile reversal learning*

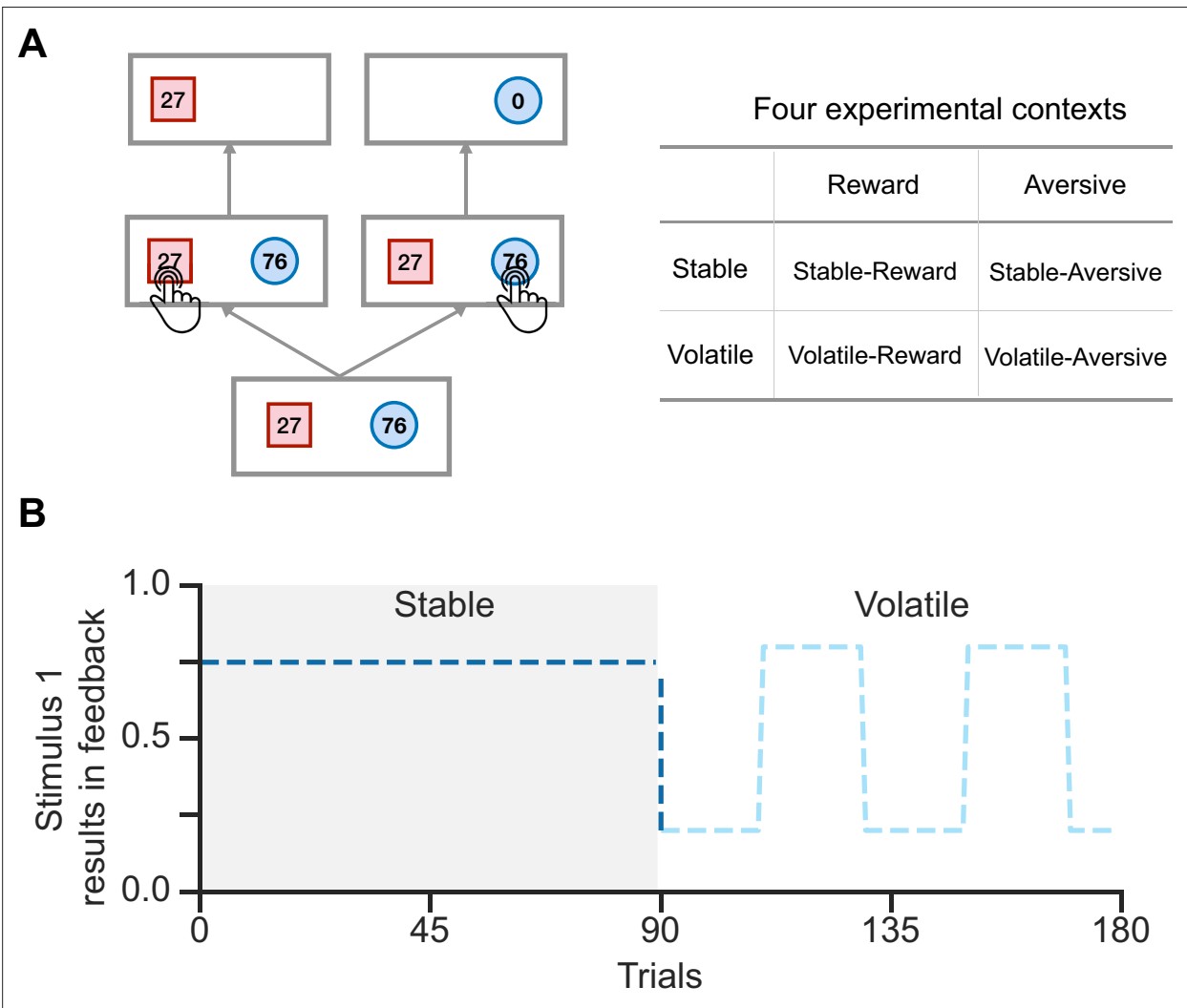

**Figure 1.** Schematic diagram of the experimental task in *Gagne et al., 2020*. (**A**) In each trial, participants were presented with two stimuli associated with their potential feedback magnitude. They were instructed to choose one of the two stimuli to receive feedback, but only one stimulus would result in feedback. Participants were required to complete tasks across four experimental contexts. (**B**) Each run consisted of 90 trials in the stable context and 90 trials in the volatile context. In the stable context, the true environmental probability remains unchanged, while in the volatile context, the probability flips every 20 trials.

task (*Behrens et al., 2007*). Unlike the traditional probabilistic reversal learning task where the reward probabilities of two options switch only once (*Cools et al., 2002*), this paradigm includes two volatility conditions (see *Figure 1B*): the reward probabilities of the two options remain constant in one condition (i.e. the *stable* condition) and switch periodically in the other (i.e. the *volatile* condition).

Previous studies have often summarized human behaviors in this paradigm using the parameter of *learning rate*, which describes the efficiency with which current information is used to promote learning. These studies typically fit a specific learning rate to each context, resulting in different learning rate values for different contexts. Using this method, previous studies have reached two important conclusions. First, human participants are able to flexibly adapt to changes in environmental volatility, as evidenced by increasing and decreasing the learning rate in response to volatile and stable conditions. This observation is often referred to as the *learning rate adaptation* effect. Second, individuals with several psychiatric disorders, including anxiety and depression, have been found to have a reduced ability to adapt their learning rate in response to environmental volatility (*Behrens et al., 2007*; *Browning et al., 2015*; *Gagne et al., 2020*). This hallmark can also be indicative of atypical behaviors (*Browning et al., 2015*; *Gagne et al., 2020*), psychosis (*Powers et al., 2017*), and autism spectrum disorder (*Lawson et al., 2017*).

However, the current approach to understanding human behaviors using learning rates has two main limitations. First, the traditional approach increases the number of learning rates as the number of contexts increases, thereby increasing the risk of overfitting. Second, this approach implicitly assumes that learning rate differences can account for all behavioral differences between stable/volatile rewarding contexts and group differences between healthy controls and patients with psychiatric disorders. However, the learning rate is not directly observable and is often estimated by model fitting, which limits its interpretability. The goal of this work is to offer an alternative explanation for human learning behaviors in volatile reversal learning tasks, moving beyond the traditional focus on learning rates. We hypothesize that the differences between stable/volatile contexts and between healthy/patient groups mainly arise from preferences for different decision strategies (*Daw et al., 2011*; *Fan et al., 2023*). We, therefore, constructed a novel MOS model, which postulates that an observer makes decisions by combining three strategies that balance reward and cognitive resources (*Gershman et al., 2015*; *Griffiths et al., 2015*). First, we consider the most rewarding strategy, *Expected Utility* (EU), which guides decision-making based on the expected utility of each option (calculated as probability multiplied by reward magnitude) (*Von and Morgenstern, 1947*). This EU strategy yields the maximum amount of reward, but the utility calculation itself consumes considerable cognitive resources. Alternatively, humans may choose simpler strategies, e.g., the *magnitude-oriented* (MO) strategy, in which only the reward magnitude was considered during the decision process, and the *habitual* (HA) strategy, in which people simply repeat decisions frequently made in the past regardless of reward magnitude (*Wood and Rünger, 2016*). We use the preference for these decision strategies to roughly estimate participants' decision styles in the volatile reversal task.

In this study, we apply and examine the MOS model on a dataset previously reported by *Gagne et al., 2020* and demonstrate its ability to explain the impaired learning behaviors of individuals diagnosed with depression and anxiety. First, we show that depression and anxiety patients exhibit three signature behavioral patterns indicative of inferior task performance. The MOS model not only qualitatively captures all three behavioral patterns but also quantitatively provides a better fit to the behavioral data than previous models. We then revisit the classical learning rate adaptation theory and show that strategy preference readily accounts for two key learning rate adaptation effects observed in prior research. Our work presents an alternative explanation for the effects of environmental volatility on human learning and highlights the importance of understanding atypical patient behaviors through the lens of decision-making strategies rather than solely focusing on learning rate.

## Results

We examined human volatile reversal learning behaviors in a public data set reported by *Gagne et al., 2020*. In a volatile reversal learning task, participants chose between two shaped stimuli to receive feedback. Participants received the presented feedback (e.g. '27' on the 'square') when choosing the *feedback stimulus*; otherwise, they received '0' (*Figure 1A*). The task was divided into four contexts: reward or aversive feedback types crossed with stable or volatile conditions. Participants earned points, which were convertible to monetary rewards, in the reward context or received electric shocks

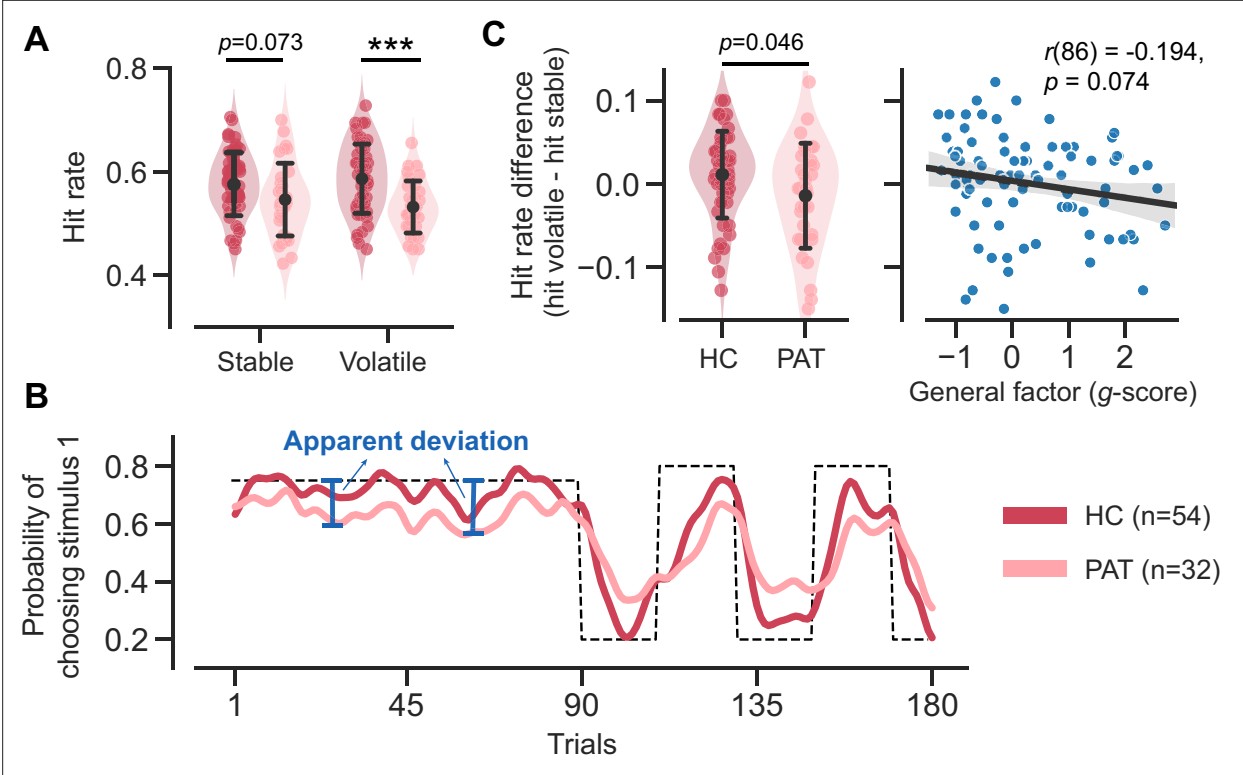

**Figure 2.** Task performance comparison between healthy control participants and patients diagnosed with major depressive disorder (MDD) and generalized anxiety disorder (GAD). Significance symbols: *p<0.05; **p<0.01; ***p<0.001; *n.s.*: non-significant. Abbreviations: HC, healthy controls; PAT, patients. (**A**) Comparison of hit rates for healthy controls and patients in stable and volatile contexts. Error bars represent the standard deviation of the estimated mean across 86 participants. (**B**) Learning curves for healthy controls and patients throughout the learning process. The dashed line represents the exemplar feedback probability sequence. For runs that do not follow this exemplar sequence (e.g. starting with volatile and then moving to stable conditions), responses were converted to match the exemplar sequence. The learning curves for both groups were then generated by averaging these converted responses across participants within each group. For better visualization, these curves were then smoothed using a Gaussian kernel with a standard deviation of two trials. The blue arrows indicate the apparent deviation between the true feedback probability and the patients' asymptotic performance. (**C**) Hit rate differences for healthy controls and patients and their relationship with participants' symptom severities. Error bars represent the standard deviation of the estimated mean across 54 healthy controls and 32 patients, respectively.

in the aversive context. In stable, one stimulus had a higher fixed feedback probability (always 75%), while in volatile, the dominant stimulus switched every 20 trials (either 20% or 80%), requiring active learning of stimulus-feedback contingencies (*Figure 1B*). Each participant was instructed to complete two runs of the volatile reversal learning task, one in the reward context and the other in the aversive context. Each run consisted of 180 trials, with 90 trials in the stable context and 90 in the volatile context (*Figure 1B*). No additional hints were provided about the transition from one context to another; therefore, participants needed to infer the current context on their own.

Eighty-six participants took part in this experiment, comprising 20patients with major depressive disorder (MDD), 12patients with generalized anxiety disorder (GAD), and 54 healthy control participants. In this article, we grouped the MDD and GAD individuals into a patient group and the remaining 54 participants into a healthy control group. Please refer to the Materials and methods section for a more detailed introduction to the methods and participant groups.

## Atypical behavioral patterns in MDD and GAD patients

Patients with MDD and GAD exhibit three key behavioral patterns as compared to healthy controls. First, patients achieved a significantly lower hit rate (averaged across stable and volatile contexts) as compared to the healthy controls (*Figure 2A*; $t(70.541) = 3.326$, p = 0.001, Cohen's d = 0.723). The hit rate refers to the accuracy of a participant in choosing the correct stimulus throughout the task. Specifically, the correct stimulus is the one that yields reward points in the reward context or avoids electric shocks in the aversive context.

Second, we observed two atypical features of learning curves in the patient group (*Figure 2B*). The patients' learning curves took more trials to converge to an asymptote (i.e. seemingly slower learning). Additionally, there was a larger apparent deviation (*Figure 2B*, blue arrows) from the true feedback probability. The apparent deviation indicates that the learning curve of the patient group could never converge to the true feedback probability, even given a sufficient number of trials in a stable context.

Third, aside from the lower learning rate and atypical learning curves that indicate inferior performance in the patient group, we further discovered a reduced hit rate difference within the patient group (*Figure 2* and *t*(55.648) = 2.038, p = 0.046, Cohen's d = 0.478). Interestingly, this hit rate difference is marginally associated with the severity of participants' symptoms (*r*(86) = –0.194, p = 0.074), as measured using the bifactor analysis reported in *Gagne et al., 2020*. This analysis decomposes symptoms into specific factors for anxiety and depression, with the *g-score* representing the common symptoms between them. The hit rate difference across volatile/stable contexts may be due to the setting of true probability (0.8 in the volatile context and 0.75 in the stable context).

## The mixture-of-strategies model captures group differences in learning behaviors

In a volatile reversal learning task, each participant in the experiment faces two fundamental challenges. First, they must engage in decision-making, constructing a policy $\pi$ to determine an action that maximizes benefit. Second, they must learn to figure out the feedback probability $\psi$ for each stimulus, which is not explicitly stated, through their interactions with the environment. To gain insights into how cognitive impairments lead to the above-mentioned atypical behaviors in the patient group, we developed four families of computational models. All models utilize the same reinforcement learning method for learning feedback probability $\psi$ but differ in how they construct their policies $\pi$ for decision-making.

Our target model family, known as MOS, posits that behavioral differences across the two participant groups and between stable/volatile contexts can be attributed to varying weightings of multiple decision strategies: EU, MO, and HA

$$\pi\left(s \mid \psi, m, \pi_{HA}\right) = w_{EU}\pi_{EU}\left(s \mid \psi, m\right) + w_{MO}\pi_{MO}\left(s \mid m\right) + w_{HA}\pi_{HA}\left(s\right)$$

This particular three-strategy configuration was chosen as the representative model because it best accounts for human behavioral data (Figure 3—figure supplement 1). The EU strategy ($\pi_{EU}$) postulates that human agents rationally calculate the value of each stimulus $s$ by multiplying its estimated feedback probability $\psi$ with reward magnitude $m$. The MO strategy ($\pi_{MO}$) only focuses on feedback magnitude $m$, disregarding feedback probability $\psi$. This is certainly an irrational strategy but more economical in terms of cognitive efforts. The HA strategy ($\pi_{HA}$) reflects the tendency to repeat previous frequent choices, depending on neither feedback magnitude $m$ nor feedback probability $\psi$. Parameters $w_{EU}$, $w_{MO}$, and $w_{HA}$ are the weighting of each strategy representing a decision-maker's preference for each strategy. We fit two MOS variants, MOS6 and MOS22. Both models have identical

**Table 1.** Model's parameters.

| Model | Context-free parameters | Context-dependent parameters |
|---|---|---|
| MOS6 | $\beta, \alpha_{HA}, \alpha_\psi, w_{EU}, w_{MO}, w_{HA}$ | |
| MOS22 | $\beta, \alpha_{HA}$ | $\alpha_{\psi+}, \alpha_{\psi-}, w_{EU}, w_{MO}, w_{HA}$ |
| FLR6 | $\alpha_{HA}, r, \beta_{HA}, \alpha_\psi, \beta, \lambda$ | |
| FLR22 | $\alpha_{HA}, r$ | $\beta_{HA}, \alpha_{\psi+}, \alpha_{\psi-}, \beta, \lambda$ |
| RS3 | $\beta, \alpha_\psi, \gamma$ | |
| RS13 | $\beta$ | $\alpha_{\psi+}, \alpha_{\psi-}, \gamma$ |
| PH4 | $\alpha_\psi^0, k, \eta, \beta$ | |
| PH17 | $\alpha_\psi^0$ | $k_+, k_-, \eta, \gamma$ |

update rules; however, MOS22, the context-dependent variant, fits a separate set of parameters to each experimental context, whereas MOS6, the context-free variant, uses one set of parameters for all contexts (*Table 1*). This approach applies to the other three model families, each offering two distinct variants.

In contrast to the mixture-of-strategies account, the Flexible-Learning-Rate (FLR) models—the context-free FLR6 and the context-dependent FLR22—hypothesize that behavioral differences between groups and contexts primarily arise from different learning rates, known as learning rate adaptation. These models, reported as the best models by *Gagne et al., 2020*, select stimuli with higher values, estimated by a linear combination of differences in feedback probability, (non-linear) feedback magnitude, and the stimuli's consistency with habitual behaviors. The Risk-Sensitive (RS) models (RS3 and RS13), adopted from *Behrens et al., 2007* and *Browning et al., 2015*, share the same hypothesis about human behavioral differences. These models use the EU strategy for decision-making and consider a subjective distortion of the learned feedback probability when calculating the expected value. To further investigate the hypothesis regarding differences in learning rates, we tested a family of models with a built-in adaptive learning rate, known as the Pearce-Hall models (PH4, PH17). See the Materials and methods section for the detailed model implementations.

The model fitting reveals that the MOS models accurately account for human behaviors. MOS6 and MOS22 were the best-fitting models in terms of the Bayesian Information Criterion (BIC; *Schwarz, 1978*) and Akaike Information Criterion (AIC; *Akaike, 1974*), respectively (*Figure 3A*). The group-level Bayesian model comparison (*Rigoux et al., 2014*) further supports MOS6 as the best-fitting model (*Figure 3B*). These model comparisons highlight that the MOS models outperform the other three families of models supporting the learning adaptation account, suggesting that behavioral variations might not be fully captured by learning rate adaptations alone.

The MOS models can not only better capture the data quantitatively, but they can also effectively reproduce the three key behavioral differences between the groups. The MOS models reproduce the lower hit rate (*Figure 3C*)**,** reduced hit rate difference (*Figure 3D*), and slower learning curves with apparent deviations (*Figure 3E*) observed in the patient group, whereas the FLR models struggle to produce all these effects. See *Figure 3—figure supplements 2–3* for the behavioral patterns for all models.

In short, we conclude that the MOS models best account for human behavioral data both qualitatively and quantitatively. In the following sections, we will analyze the fitted parameters of the MOS models to interpret the atypical behavioral patterns of the patient group.

## MDD and GAD patients favor simpler decision strategies

We first focused on the fitted parameters of the MOS6 model. We compared the weight parameters ($w_{EU}$, $w_{MO}$, $w_{HA}$) across groups and conducted statistical tests on their logits ($\lambda_{EU}$, $\lambda_{MO}$, $\lambda_{HA}$). The patient group showed a ~37% preference towards the EU strategy, which is significantly weaker than the ~50% preference in healthy controls (healthy controls' $\lambda$: M = 0.991, SD = 1.416; patients' $\lambda$: M = 0.196, SD = 1.736; $t(54.948)$ = 2.162, p = 0.035, Cohen's d = 0.509; *Figure 4A*). Meanwhile, the patients exhibited a weaker preference (~27%) for the HA strategy compared to healthy controls (~36%) (healthy controls' $\lambda$: M = 0.657, SD = 1.313; patients' $\lambda$: M = –0.162, SD = 1.561; $t(56.311)$ = 2.455, p = 0.017, Cohen's d = 0.574), but a stronger preference for the MO strategy (14% vs 36%; healthy controls' $\lambda$: M = –1.647, SD = 1.930; patients' $\lambda$: M = –0.034, SD = 2.091; $t(63.746)$ = –3.510, p = 0.001, Cohen's d = 0.801). Most importantly, we also examined the learning rate parameter in the MOS6 but found no group differences ($t(68.692)$ = 0.690, p = 0.493, Cohen's d = 0.151). These results strongly suggest that the differences in decision strategy preferences can account for the learning behaviors in the two groups without necessitating any differences in learning rate per se.

The MOS6 assumes no parameter differences across the four contexts, which may dilute the group differences in learning rate. We further analyzed the MOS22, which explicitly estimates different sets of weighting and learning rate parameters in different contexts (i.e. context-dependent), and found a consistent conclusion about participants' strategy preferences. We first conducted three separate 2 × 2 × 2 ANOVAs, each setting the logit of a weighting parameter ($\lambda_{EU}$, $\lambda_{MO}$, $\lambda_{HA}$) as the dependent variable, and participant groups (healthy control/patient) as the between-subject variable, and volatile contexts (stable/volatile), and feedback contexts (reward/aversive) as within-subject variables. We again found a weaker preference for EU ($F(1, 80)$ = 13.537, p<0.001, $\eta^2$ = 0.084) and a stronger

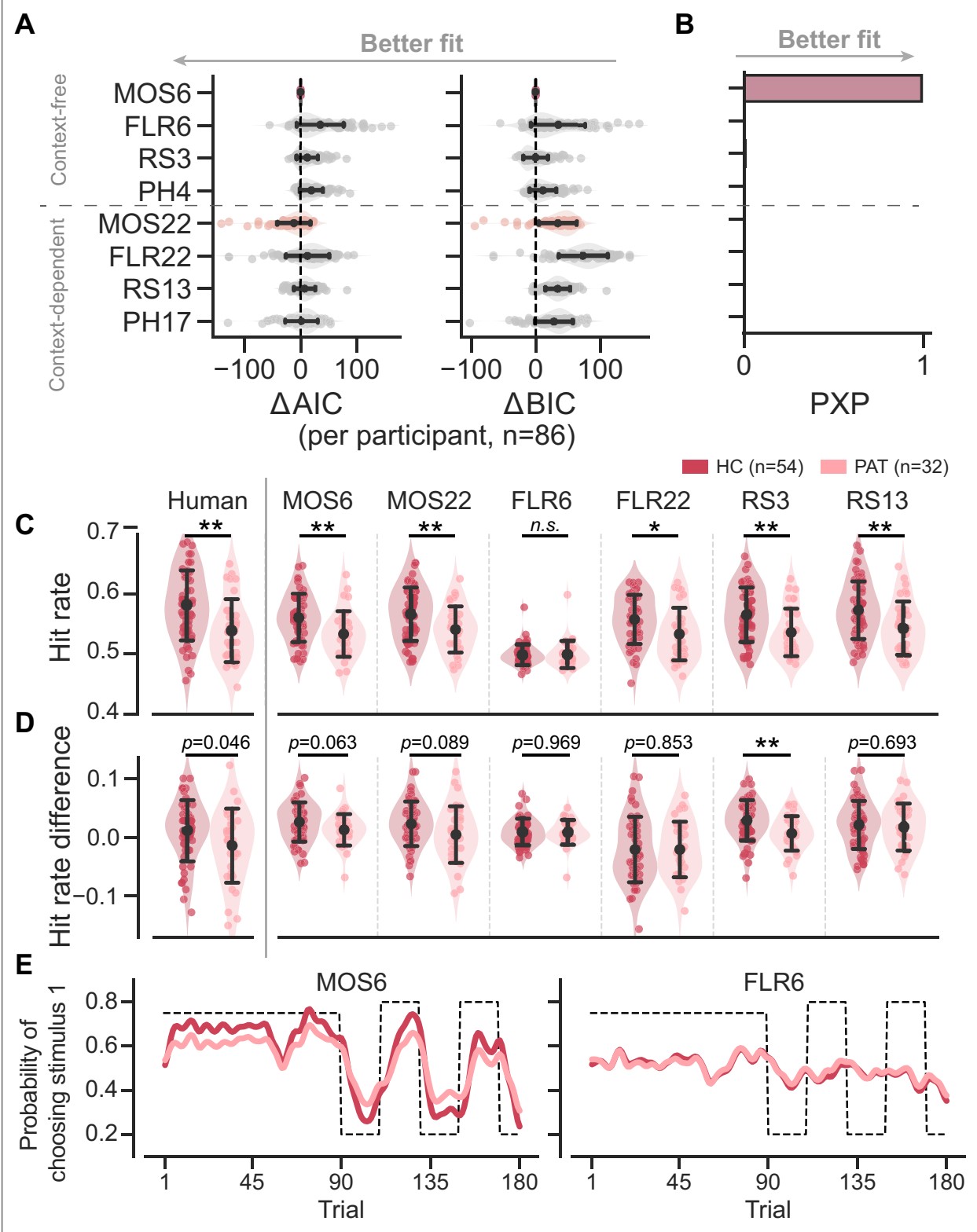

**Figure 3.** Models' quantitative and qualitative fit to human behavioral data. Significance symbols: *p<0.05; **p<0.01; ***p<0.001; *n.s.*: non-significant. Abbreviations: HC, healthy controls; PAT, patients. (**A**) Relative performance of models compared to the MOS6 model, as measured by the Akaike Information Criterion (AIC), and Bayesian Information Criterion (BIC). Each dot represents a model's fit for an individual participant, with error bars showing the standard deviation of the estimated mean across 86 participants. (**B**) Group-level Bayesian model selection as indicated by Protected

*Figure 3 continued on next page*

*Figure 3 continued*

Exceedance Probability (PXP). (**C–E**) Models' predicted hit rate (**C**) hit rate differences (**D**) and learning curves (**E**) for healthy controls and patients, respectively. Error bars denote the standard deviation of the estimated mean across 54 healthy controls and 32 patients, respectively.

The online version of this article includes the following figure supplement(s) for figure 3:

**Figure supplement 1.** The fit to the human data of different mixture-of-strategies (MOS) variants.

**Figure supplement 2.** Hit rates (**A**) and hit rate differences (**B**) for all models.

**Figure supplement 3.** Simulated learning curves for the healthy control (HC) and patient (PAT) groups, each averaged from 100 simulations within the group and were smoothed with a Gaussian kernel (standard deviation of two trials).

preference for MO ($F(1, 80) = 7.791$, p = 0.009, $\eta^2 = 0.046$) in the patient group (***Figure 4—figure supplement 1A***). However, unlike the MOS6, the MOS22 revealed no significant group difference was observed in the HA strategy ($F(1, 80) = 0.020$, p = 0.887, $\eta^2 < 0.001$), and a significantly a stronger preference for EU under the reward context Bonferroni-$t$ = 2.243, p = 0.028, Cohen's d = 0.209. This suggests a possible confounding between the EU and HA strategies. Next, we examined the learning rates of the MOS22. A $2 \times 2 \times 2 \times 2$ ANOVA was performed with the (log) learning rate parameter as the dependent variable, outcome valence (better/worse than expectation) as a within-subject variable in addition to the three independent variables (group/volatile context/feedback context) as introduced above. We again found no significant difference between patients and healthy controls ($F(1, 77) = 0.393$, p = 0.533, $\eta^2 = 0.003$; ***Figure 4—figure supplement 1B***). Most importantly, the MOS22 model revealed no learning rate adaptation effect, as indicated by the learning rate parameters in the volatile context not being significantly larger than that in the stable context ($F(1, 77) = 0.126$, p = 0.724, $\eta^2 < 0.001$; ***Figure 4—figure supplement 1C***). Based on these findings, we drew two conclusions. First, MOS6 and MOS22 made consistent descriptions of participants' strategy preferences during decisions: the behavioral differences between the two participant groups were mainly attributed to differences in their strategy preferences, rather than their learning rates. Second, the learning rate adaptation effect may be simply explained by context-free strategy preferences. We will further explain this second point in later sections.

## Understanding patients' inferior task performances through strategy preferences

In this section, we illustrate how strategy preferences account for the three learning behavioral differences observed between the two participant groups, as shown in ***Figure 2***. To better understand how each decision strategy influences the three behavioral patterns, we simulated the MOS6 model using the median fitted parameters and outputted the decisions for each strategy (see Simulation details in Materials and methods).

For hit rate, our simulations showed that the EU strategy achieved the highest hit rate, while the MO strategy basically performed at the chancel-level (***Figure 4B***). These results are intuitively understandable. Since the hit rate is defined based on feedback probability, the EU strategy, which actively tracks this probability, should be able to achieve a high hit rate. In contrast, the MO strategy, which completely ignores feedback probability, should achieve a chance-level hit rate. Interestingly, our simulations also showed that the HA strategy achieved an above-chance hit rate. This is because, although the HA strategy appears not to consider feedback probability directly, it still somewhat tracks feedback probability by simply repeating the past choices made by the EU. Accordingly, assigning lower weights to the two higher-hit-rate strategies, EU and HA (i.e. higher weighting on MO), naturally leads to inferior performance in the patients (***Figure 2A***).

We also visualized the simulated learning curves for each strategy (averaged across the two groups) throughout the task (***Figure 4C***). In both stable and volatile contexts, the EU strategy quickly approximates and converges to the true feedback probability. The HA strategy takes more trials to approach the true feedback probability, exhibiting slower learning. The MO strategy does not respond to environmental feedback, resulting in an almost flat learning curve. When the learning curves are combined separately for the two groups, we recover the seemingly slower learning curve in the patient group due to their stronger preference for the MO strategy (***Figure 4E***). We also noted the larger apparent deviation from the true feedback probability in the patient group. These two features in ***Figure 2B***

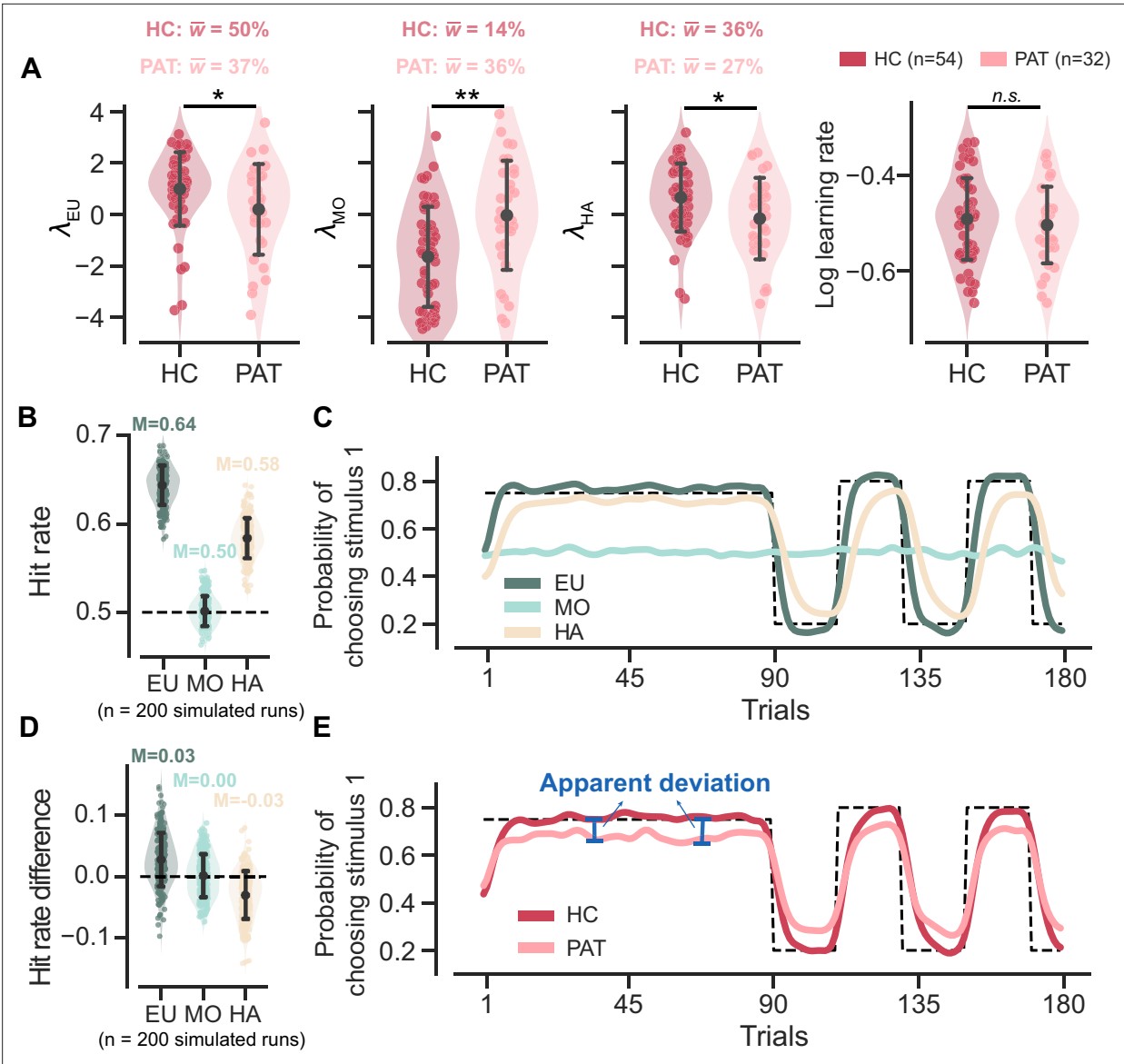

**Figure 4.** Parameter analyses of the MOS6 model and simulated behaviors for all three strategies. Significance symbol conventions are: *p<0.05; **p<0.01; ***p<0.001; *n.s.*: non-significant. Abbreviations: HC, healthy controls; PAT, patients. (**A**) The fitted weighting parameters and learning rate of the MOS6 model. The y-axis means averaged preference over different volatile contexts (volatile/stable) and feedback contexts (reward/aversive). $\bar{w}$ indicates the averaged weighting parameters for each participant group. Error bars denote the standard deviation of the estimated mean across 54 healthy controls and 32 patients, respectively. (**B**) Simulated hit rates for the three decision strategies. Error bars represent the standard deviation across 200 simulations. The 200 simulations were evenly divided between groups using parameters similar to the healthy control group and the patient group. The groups differed only in their strategy preference (differences in $w_{EU}, w_{MO}, w_{HA}$) while all other parameters remained constant. For more simulation details, refer to Materials and methods, Simulation details. (**C**) The average simulated learning curve for each strategy across 200 simulations, was smoothed with a Gaussian kernel (standard deviation of two trials). (**D**) Simulated hit rate differences between volatile and stable for the three decision strategies. Error bars represent the standard deviation across 200 simulations. (**E**) Simulated learning curves for the healthy controls and patients, each averaged from 100 simulations within the group and smoothed with a Gaussian kernel (standard deviation of two trials).

The online version of this article includes the following figure supplement(s) for figure 4:

**Figure supplement 1.** Parameter analyses of the MOS22 model.

can thus be readily explained by the patients' stronger preference for the MO strategy, as the MO strategy does not learn feedback probability at all and exhibits a flat learning curve.

For the hit rate differences between the stable and volatile contexts, our simulations showed that the EU strategy achieves a higher hit rate in the volatile context than in the stable context (i.e. positive

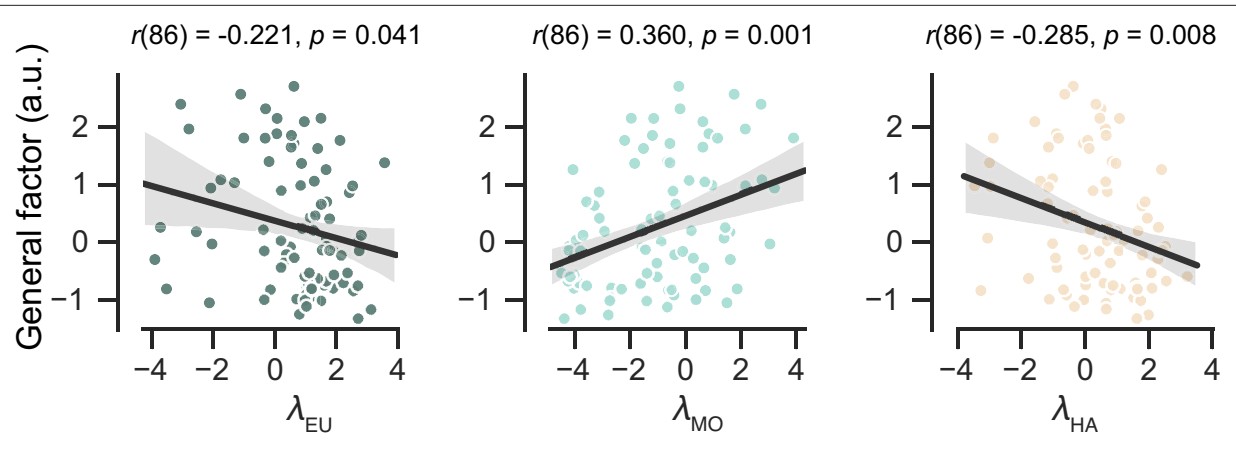

**Figure 5.** Predict participants' symptom severity (*g* score) using strategy preferences of the MOS6 model. Each dot represents one participant. The shaded areas reflect 95% confidence intervals of the regression prediction.

The online version of this article includes the following figure supplement(s) for figure 5:

**Figure supplement 1.** Strategy preferences predict participants' general factor score (*g* score) in the bifactor analysis reported by *Gagne et al., 2020*.

hit rate difference) (*Figure 4D*). This is attributed to the EU strategy's active tracking of feedback probability (i.e. the maximum possible hit rate), which increases from 75% in the stable context to 80% in the volatile context. Conversely, there were no changes in the MO strategy's hit rate from the stable to volatile contexts (i.e. 0 hit rate difference) because MO does not track feedback probability. Additionally, we found that the hit rate of the HA strategy was higher in the stable context than in the volatile (i.e. negative hit rate difference). This is possibly because the HA strategy requires more time to relearn true probability (*Figure 4C*), particularly in the volatile context where the true probability frequently flips. Based on this, we can roughly estimate the hit rate difference for the healthy control group as~0.042 ($\bar{w}_{EU}^{HC} \times 0.3 + \bar{w}_{MO}^{HC} \times 0 + \bar{w}_{HA}^{HC} \times -0.3 = 0.5 \times 0.3 + 0 + 0.36 \times -0.3$) and for the patient group as~0.030 ($\bar{w}_{EU}^{PAT} \times 0.3 + \bar{w}_{MO}^{PAT} \times 0 + \bar{w}_{HA}^{PAT} \times -0.3 = 0.37 \times 0.3 + 0 + .27 \times -0.3$). This explains why healthy controls exhibited slightly a larger hit rate difference than the patient participants (*Figure 2C*).

### Atypical strategy preferences are connected to the general severity of anxiety and depression

We investigated the relationship between strategy preferences in the MOS6 model and symptom severity in the patient group (*Figure 5*). Our findings indicate that patients with severe symptoms exhibit a weaker preference for the cognitively demanding EU strategy (Pearson's *r* = –0.221, p = 0.040) and a stronger preference for the simpler MO strategy (Pearson's *r* = 0.360, p = 0.001). Additionally, there was a significant correlation between symptom severity and the preference for the HA strategy (Pearson's *r* = –0.285, p = 0.007). These results highlight the strong clinical relevance of strategy preferences.

For completeness, we examined the correlation between learning rate adaptation (log volatile learning rate – log stable learning rate) and symptom severity within the MOS22 model (*Figure 5—figure supplement 1*). Not surprisingly, we found no significant correlation (*r*(86) = 0.130, p = 0.233), which is consistent with our finding of no difference in learning rates across the two volatile contexts.

### Strategy preferences may explain the learning rate differences across groups and contexts

Previous studies using probabilistic reversal learning tasks have made three major conclusions about learning rate. First, it has been documented that individuals with anxiety and depression have a smaller learning rate parameter (*Chen et al., 2015*; *Pike and Robinson, 2022*), thereby exhibiting a slower learning curve (*Figure 2C*) and, possibly, a lower hit rate (*Figure 2A*). Second, human participants have been found to be able to flexibly increase their learning rate in response to high environmental volatility (*Behrens et al., 2007*). Third, patient participants may exhibit a deficit in such

learning rate adaptation (*Browning et al., 2015*; *Gagne et al., 2020*), exhibiting a lesser extent of increase (*Figure 2B*).

However, we recognize two limitations in this learning rate interpretation. First, a higher learning rate does not necessarily improve the hit rate; it may lead to overreacting to feedback from stimuli with low probabilities. Second, and more importantly, a reduced learning rate merely prolongs the time needed to approach the true probability (*Boyd and Vandenberghe, 2004*) but cannot explain the apparent deviation from the true probability observed in patient participants (*Figure 2B*). In contrast, a mixture of strategies can naturally account for these two phenomena. As mentioned in the previous section, the mixture of EU and MO results in both a seemingly lower learning curve and a larger apparent deviation.

Here, we further demonstrate that the behavioral differences caused by a mixture of strategies could reflect the learning rate adaptation across the stable and volatile contexts. We used the MOS6 to synthesize behavioral data for agents resembling healthy controls and patients by controlling all parameters except for the decision weights. Specifically, we set the weights of $w_{EU}$ to 60%, $w_{MO}$ to 15% and $w_{HA}$ to 25% for the healthy control group, and the weights of $w_{EU}$ to 15%, $w_{MO}$ to 60% and $w_{HA}$ to 25% for the patient group, with all other parameters fixed to the median values across all participants (see more details in Materials and methods). We simulated each group 20 times, and the simulated data reproduced the slower learning curve and the apparent difference in the patient group (*Figure 6A*). We fit the simulated data generated by MOS6 with the FLR22 model and found significant differences in the (log) learning rate between stable and volatile contexts (paired $t$-test(39) = –3.217, p = 0.003, Cohen's d = 0.721; *Figure 6B*). Furthermore, the agent resembling the patient group demonstrated a trend toward reduced learning rate adaptation compared to the agent resembling the healthy control group (*Figure 6C*), consistent with the learning rate adaptation theory. These findings suggest that what might be perceived as learning rate adaptation could result from a mixture of strategy preferences. This observation also implies that strategy preferences may, at least partially, explain the maladaptive adaptations in learning rate observed in patients in response to environmental volatility.

## Model and parameter recovery analyses support model and parameter identifiability in MOS

Although we have previously demonstrated that the MOS models are quantitatively best-fitting, there are two potential confounding factors. First, it is possible that differences in learning rate, rather than differences in strategy preference, could produce the same behavioral outcomes that are indistinguishable by the model fitting. If this holds, the MOS model might be problematic, as all learning rate differences may be automatically attributed to strategy preferences because of some unknown idiosyncratic model fitting mechanisms. Second, the fact that the MOS models outperform the others may be partly due to an unknown bias in the model design. It is possible that the MOS models always win, irrespective of how the data is generated.

To circumvent these issues, we conducted parameter recovery analyses on MOS6 and model recovery analyses on all models to investigate the identifiability of true parameters and models. For parameter recovery, we generated 80 synthetic datasets using 80 different parameter sets, each varying the four parameters of interest $\{\alpha_\psi, \lambda_{EU}, \lambda_{MO}, \lambda_{HA}\}$, with the remaining parameters fixed to the median of the fitted parameters ($\beta$ = 10.803, $\alpha_{HA}$ = 0.423). Each synthetic dataset consisted of 10 runs, resulting in a total of 800 synthetic runs (80 parameter sets × 10 runs). For each dataset, we fitted our MOS6 model and compared the fitted parameters to the ground-truth parameters. The parameter recovery results (*Figure 7A*) demonstrate that the true parameters can be accurately estimated and identified (all Pearson's $r$s >0.720), indicating that the effects of learning rate and weighting parameters are not interchangeable in the MOS6 model.

For model recovery, we sampled 40 participants (20 in each group) and used their fitted model parameters to generate synthetic datasets from each of the eight models. For each participant, we simulated 10 runs of behavioral data, resulting in a total of 3200 synthetic runs (8 generating models × 40 participants × 10 runs). We then fit all eight models to every dataset using the MAP method as the same before. The best-fitting model was always the one that generated the data, as indicated by all three quantitative metrics: AIC, BIC, and PXP (*Figure 7B*). We note that the RS3 and PH4 models tend to account well for each other. The MOS6 model achieves good fitting performances on synthetic

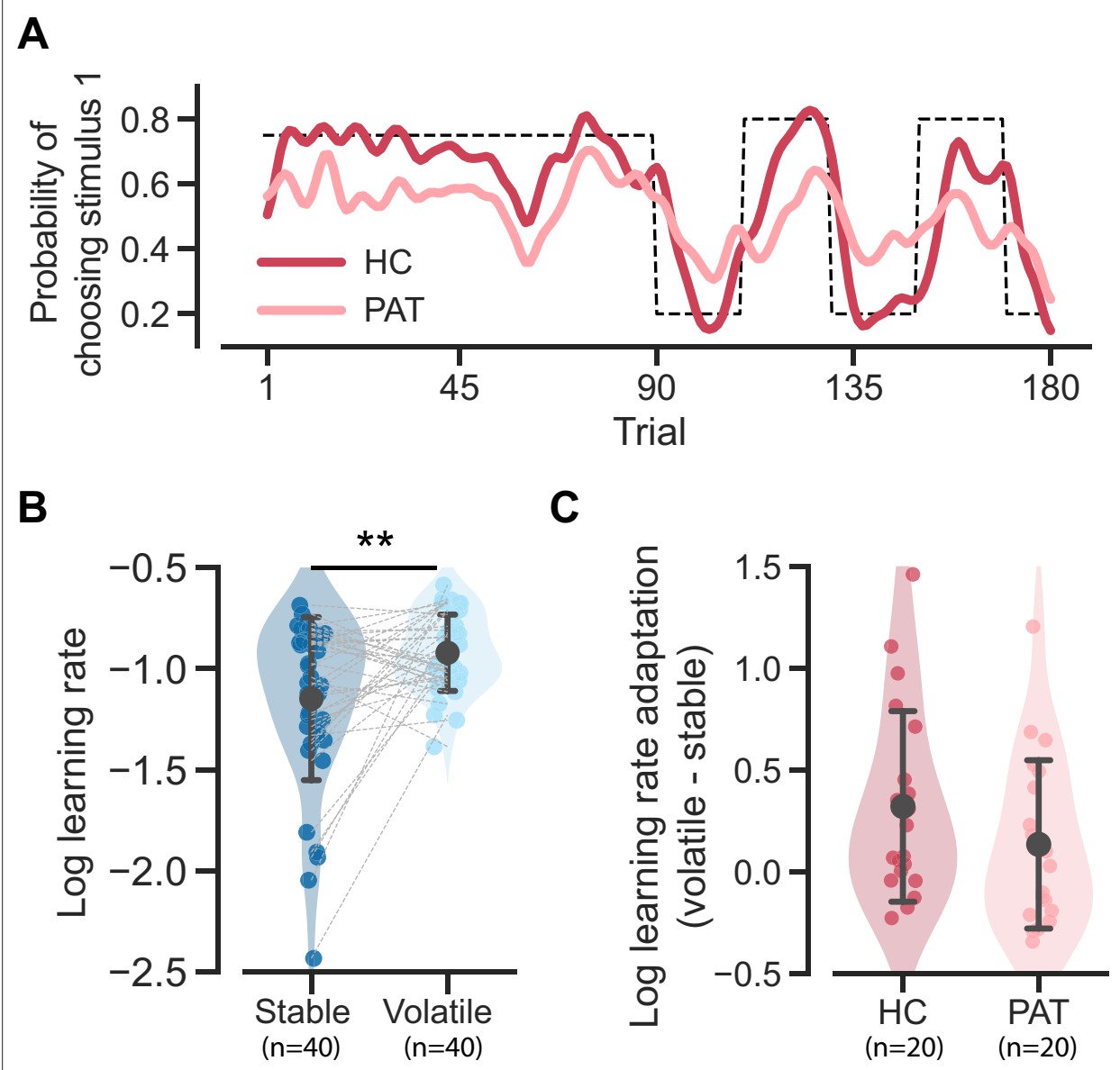

**Figure 6.** Reproduction of the two learning rate adaptation effects using the MOS6 model. Significance symbol conventions are: *p<0.05; **p<0.01; ***p<0.001; *n.s.*: non-significant. HC represents the healthy-control-like agent; PAT represents the patient-like agent. (**A**) Simulated learning curves for the healthy controls and patients generated by the MOS6 model. Both curves are averaged over 80 runs of tasks (4 task sequences × 20 experiments) and are smoothed with a Gaussian kernel (standard deviation of two trials). (**B**) The fitted FLR22 learning rate parameters are for the stable context and the volatile context. Error bars denote the standard deviation across 40 synthesized datasets. (**C**) Learning rate adaptations, calculated by log volatile learning rate – log stable learning rate, for the healthy control-like agent and for the patient-like agent. Error bars stand for the standard deviation across 20 synthesized datasets.

datasets generated by the RS3 and PH4 models, but not vice versa. The slight confusion between RS3, PH4, and MOS6 is because they all include the EU strategy. Most importantly, the MOS and FLR models cannot adequately account for each other's synthetic datasets, strongly supporting the independent computational effects of strategy preference and learning rate.

## Discussion

In this article, we propose to understand humans' learning behaviors, especially the differences between healthy controls and patients, in the volatile reversal learning task through the lens of a

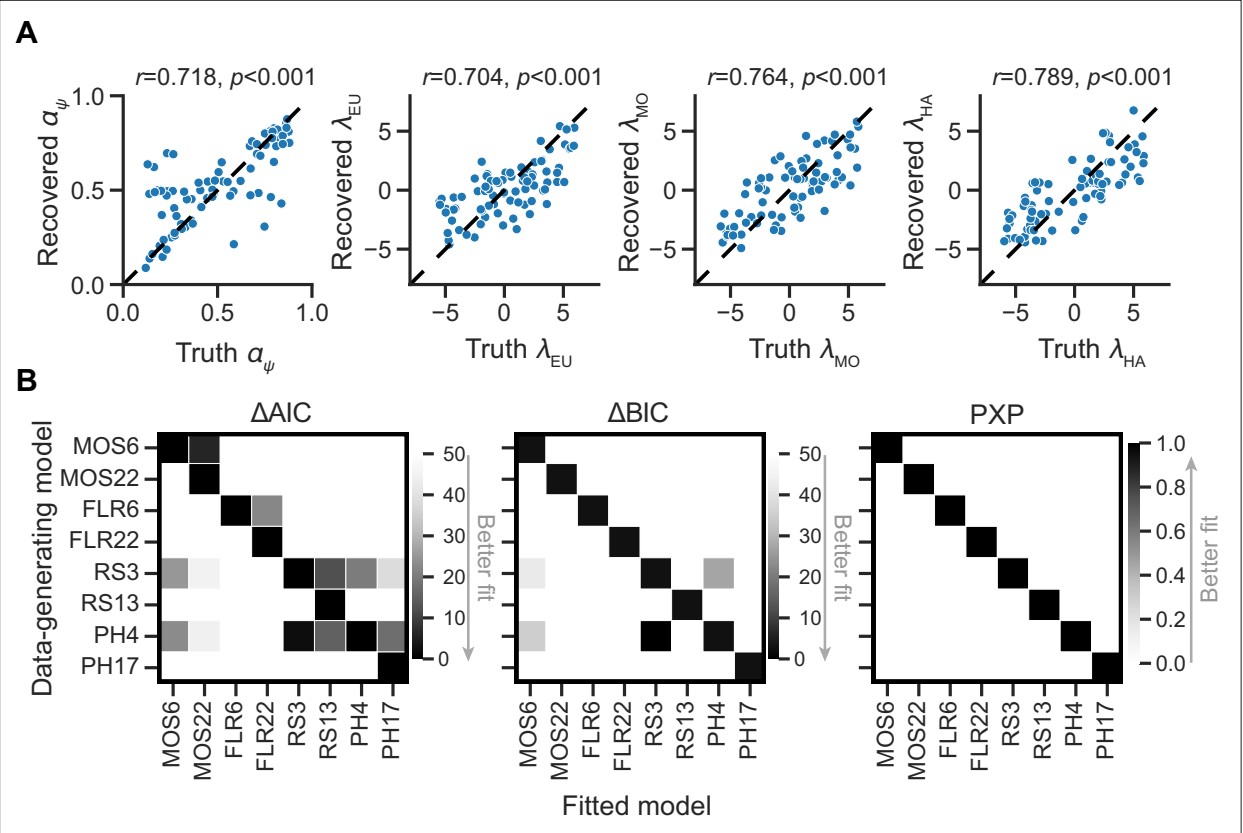

**Figure 7.** Parameter and model recovery analyses. (**A**) Parameterrecovery for the MOS6 model. (**B**) Model recovery analysis, showing the performance of models as evaluated by averaged relative Akaike Information Criterion (AIC) and Bayesian Information Criterion (BIC), as well as Protected Exceedance Probability (PXP) scores for synthesized data generated from each of the eight models. Darker tiles indicate better fits to the synthesized data.

mixture-of-strategies. We develop the MOS model, which assumes that human participants make decisions by combining three distinct components: the EU, MO, and HA strategies. The EU strategy is rewarding but cognitively demanding, in contrast to the other two strategies, which are simpler and heuristic but less rewarding. We show that the MOS model can qualitatively capture several behavioral patterns that cannot be explained by previous models and quantitatively better capture human behaviors and healthy-patient differences when applied to a public dataset. The model reveals that individuals with MDD and GAD exhibit an atypical preference for simpler and less rewarding strategies (i.e. a stronger preference for the MO strategy), and this preference alone could explain their inferior task performance relative to healthy controls, as indicated by lower hit rates, reduced adaptation volatility, and slower learning curves. Furthermore, we demonstrate that the MOS model can reproduce the human behavioral learning rate adaptation effect without changing the learning rate itself. These findings suggest that a mixture of strategies provides an effective and parsimonious explanation for human learning behaviors in volatile reversal tasks.

## The role of the HA strategy in volatile reversal learning

Although many observed behavioral differences can be explained by a shift in preference from the EU to the MO strategy among patients, we also explore the potential effects of the HA strategy. Compared to the MO, the HA strategy also saves cognitive resources but yields a significantly higher hit rate (*Figure 4A*). Therefore, a preference for the HA over the MO strategy may reflect a more sophisticated balance between reward and complexity within an agent (*Gershman, 2020*): when healthier participants exhaust their cognitive resources for the EU strategy, they may cleverly resort to the HA strategy, adopting a simpler strategy but still achieving a certain level of hit rate. This explains the stronger preference for the HA strategy in the HC group (*Figure 3A*) and the negative correlation between HA preferences and symptom severity (*Figure 5*). Apart from shedding light on the cognitive

impairments of patients, the inclusion of the HA strategy significantly enhances the model's fit to human behaviors (see examples in *Daw et al., 2011*; *Gershman, 2020*; and also *Figure 3—figure supplement 1*).

## Disassociate the learning rate adaptation and mixture of strategies

It is well-established that humans apply flexible learning rates in response to environmental volatility, exemplifying the successful application of ideal observer analysis. *Behrens et al., 2007* constructed a hierarchical ideal Bayesian observer for the volatile reversal learning task that dynamically models how higher-order environmental volatility influences the updating speed of lower-order feedback probabilities. This model suggests that the human brain estimates environmental volatility, and humans are expected to exhibit a faster-updating speed for feedback probabilities in volatile contexts. Consistent with their results, the context-dependent RS13 model revealed higher learning rate parameters in volatile contexts. *Browning et al., 2015* identified the increase in learning rate from stable to volatile contexts as a hallmark of human sensitivity to environmental volatility. They found that individuals with high trait anxiety showed reduced adaptations, thus indicating lower sensitivity to volatility. *Gagne et al., 2020* extended this research to MDD and GAD patients receiving either reward or aversive feedback. Furthermore, the phenomenon of an increased learning rate from stable to volatile conditions has also been observed in other paradigms, such as the *Predictive Inference task* (*Nassar et al., 2016*; *Nassar et al., 2010*), where participants explicitly report their estimation of environmental statistics, allowing for a direct estimation of the learning rate.

Based on our findings, we applied the MOS model—an alternative but sufficiently accurate model—to the data collected from the volatile reversal task and found that the expected learning rate adaptation was not observed. Instead, the MOS model points to an alternative explanation that accounts for multiple human behavioral patterns and their symptom severity in a more parsimonious manner, involving fewer parameters. More importantly, it is possible for the MOS model to capture the pattern of learning rate adaptation without necessitating actual changes in learning rates across different contexts. These findings indicate that future studies may systematically compare the accounts of learning rate adaptation and a mixture-of-strategies.

It is important to note that learning rate adaptations and strategy preferences could simultaneously influence behaviors. However, accurately describing both mechanisms, particularly in terms of differences between populations, requires more refined behavioral paradigms. For example, it would be helpful to use paradigms like the predictive inference task (*Nassar et al., 2016*; *Nassar et al., 2010*), which allows participants to directly report their learning rates, to minimize the confounding factors in the decision process.

## Atypical learning speed in psychiatric diseases

In the present work, we found that individuals with depression and anxiety display apparent flatter learning curves in the probabilistic learning tasks (shown in *Figure 4A*). We attributed this observation to participants' strategy preferences. However, in conventional Rescorla-Wagner modeling, learning speed is primarily indicated by the learning rate parameter. For example, *Chen et al., 2015* conducted a systematic review of reinforcement learning in patients with depression and identified 10 out of 11 behavioral datasets showing either comparable or slower learning rates in depressive patients. Nonetheless, depressive patients may not always exhibit slower learning rates. In a recent meta-analysis summarizing 27 articles with 3085 participants, including 1242 with depression and/or anxiety, *Pike and Robinson, 2022* found a reduced reward but enhanced aversive learning rate. This finding yields two practical implications. First, the heterogeneous findings in the literature may arise from heterogeneous pathologies in depression and anxiety. Second, some behavioral variations introduced by strategy preferences might have been misidentified as learning rate effects. The MOS model may provide useful complementary explanations for the consequences of a spectrum of symptoms.

## Limitations and future directions

The MOS model was developed to provide context-free interpretations of the learning rate differences observed between stable and volatile contexts, as well as between healthy individuals and patients. However, we also recognize that the MOS account may not justify other learning rate effects based solely on strategy preferences. One such example is valence-specific learning rate differences, where

learning rates for better-than-expected outcomes are higher than those for worse-than-expected outcomes (*Gagne et al., 2020*). When fitted to the behavioral data, the context-dependent MOS22 model does not reveal valence-specific learning rates (*Figure 4—figure supplement 1D*). Moreover, the valence-specific effect was not replicated in the FLR22 model when fitted to the synthesized data of MOS6.

The context-dependent MOS22 model revealed several weak interaction effects, suggesting an interaction between learning adaptation and strategy preferences. For example, patients with MDD and GAD may find it too taxing to increase their learning rates in the volatile context and instead resort to simpler strategies, such as MO, as a compromise. Investigating this hypothesis may require a paradigm incorporating a self-reporting learning rate module, like the predictive inference task (*Nassar et al., 2010*), in volatile reversal learning tasks.

Theories suggest that humans increase learning rates in the volatile context due to increased perceived uncertainty about environmental statistics (*Behrens et al., 2007*; *Nassar et al., 2010*), while others propose that strategies enabling more exploration are preferred when managing uncertainty (*Fan et al., 2023*; *Wilson et al., 2014*). To explore these ideas, we may need to adjust the paradigm to offer a wider choice of stimuli, from two to three or four (i.e. set size effects). Another question is why individuals with depression and anxiety tend toward simpler decision-making strategies. *Rumination*, a maladaptive emotion regulation behavior characterized by persistent negative thoughts observed in individuals with depression (*Song et al., 2022*; *Yan et al., 2022*), may consume cognitive resources, hindering the use of the more complex but rewarding EU strategy.

## Materials and methods

In this section, we provide the mathematical and implementational details of our model. Code is available at https://github.com/fangzefunny/policy-analysis, (copy archived at *Fang, 2024*).

### Datasets

We focused on the data from Experiment 1 reported by *Gagne et al., 2020*. The data is publicly available via https://osf.io/8mzuj/. The original study included data from two experiments. The data from Experiment 2 was not used here because it was implemented on Amazon's Mechanical Turk with no information about the participants' clinical diagnoses. Here, we provide critical information about Experiment 1 (also see *Gagne et al., 2020* for more technical details).

#### Participants

Eighty-six participants took part in this experiment. The pool includes 20 patients with a major depressive disorder (MDD), 12 patients with a generalized anxiety disorder (GAD), and 54 healthy control participants. The diagnosis was made through a phone screen, an in-person screening session, and the structured clinical interview following DSM-IV-TR (SCID) in 20 MDD patients, 12 GAD patients, and 20 healthy control participants. The remaining 30 healthy control participants were recruited without SCID. In this article, we grouped the MDD and GAD individuals into a patient (PAT) group and the remaining 54 participants into a healthy control (HC) group. The detailed difference between MDD and GAD is not the focus of this paper. We will show later that the general factor behind MDD and GAD is the only factor that predicts learning behavior (see next section for details), a similar result reported in the original study (*Gagne et al., 2020*).

#### Clinical measures

The severity of anxiety and depression in all participants was measured by several standard clinical questionnaires, including the Spielberger State-Trait Anxiety Inventory (STAI form Y; *Spielberger et al., 1983*), the Beck Depression Inventory (BDI; *Beck et al., 1961*), the Mood and Anxiety Symptoms Questionnaire (MASQ; *Clark and Watson, 1991*; *Watson and Clark, 1991*), the Penn State Worry Questionnaire (*Meyer et al., 1990*), the Center for Epidemiologic Studies Depression Scale (CESD; *Radloff, 1977*), and the Eysenck Personality Questionnaire (EPQ; *Eysenck and Eysenck, 1975*). An exploratory bifactor analysis was then applied to item-level responses in all questionnaires to disentangle the variance that is common to GAD and MDD or unique to each. The results of this analysis summarized participants' symptoms into three orthogonal factors: a general factor (*g*) explaining the

common symptoms, a depression-specific factor (*f1*), and an anxiety-specific factor (*f2*), which are all included in the public dataset. Similar to the original study, here we focused on the general factor (*g* score) to indicate the participants' severity of their psychiatric symptoms.

## Stimuli and behavioral task

In a volatile reversal learning task, participants were instructed on each trial to choose between two stimuli, represented by different shapes, in order to receive feedback. The locations of the two shapes were counterbalanced across trials. The potential amount of feedback (referred to as feedback magnitude) was presented together with the stimuli. Only one of the two stimuli was associated with actual feedback (0 for the other one). The feedback magnitude, ranged between 1–99, was sampled uniformly and independently for each shape from trial to trial. Actual feedback was delivered only if the stimulus associated with feedback was chosen; otherwise, a number '0' was displayed on the screen, signifying that the chosen stimulus returned no reward.

Participants was supposed to complete this learning and decision-making task in four experimental contexts, two feedback contexts (reward or aversive) × two volatility contexts (stable or volatile). Participants received points in the reward context and an electric shock in the aversive context. The reward points in the reward context were converted into a monetary bonus by the end of the task, ranging from £0 to £10. In the stable context, the dominant stimulus (i.e. a certain stimulus induces the feedback with a higher probability) provided a feedback with a fixed probability of 0.75, while the other one yielded a feedback with a probability of 0.25. In the volatile context, the dominant stimulus's feedback probability was 0.8, but the dominant stimulus switched between the two every 20 trials. Hence, this design required participants to actively learn and infer the changing stimulus-feedback contingency in the volatile context.

Each participant was instructed to complete two runs of the volatile reversal learning task, one in the reward context and the other in the aversive context. Each run consisted of 180 trials, with 90 trials in the stable context and 90 in the volatile context. No additional hints were provided about the transition from one context to another; therefore, participants need to infer the current context on their own. A total of 79 participants completed tasks in both feedback contexts. Four participants only completed the task in the reward context, while three participants only completed the aversive task.

## Computational modeling

We first introduce our notation system. We denote each stimulus $s$ as one of two possible states $s \in \{s_1, s_2\}$. The labeled feedback magnitude (i.e. reward points or shock intensity) of the stimulus is $m(s)$, and the feedback probability is $\psi(s)$. Following the convention in reinforcement learning (*Sutton and Barto, 2018*), we presume that the decision is made from a policy $\pi$ that maps the observed magnitudes $m$ and currently maintained feedback probabilities $\psi$ to a distribution over stimuli, $\pi(s \mid m, \psi)$.

In a volatile reversal learning task, each participant in the experiment must resolve two fundamental challenges: (1) decision-making, determining an action to maximize benefit; and (2) learning, figuring out the untold feedback probability via their interaction with the environment. Here, we introduce four families of models that all utilize the same reinforcement learning method for learning feedback probability but differ in how they construct their policies for decision-making. First, the MOS model, the target model proposed in this paper, utilizes a decision-making policy consisting of a mixture of three strategies: EU, MO, and HA. Second, the FLR model, reported as the best model by *Gagne et al., 2020*, selects stimuli with higher values. The stimulus value was estimated by a linear combination of differences in feedback probability, (non-linear) feedback magnitude, and the stimuli's consistency with habitual behaviors. Third, the Risk-Sensitive (RS) model, adopted from *Behrens et al., 2007* and *Browning et al., 2015*, utilizes the EU strategy in decision-making and considers a subjective distortion of the learned feedback probability when calculating the expected value. Finally, the Pearce-Hall (PH) model, equipped with a built-in learning rate adaptation mechanism, utilizes the EU strategy for decision-making.

Notably, the MOS model, which is the core contribution of this study, posits that behavioral differences across the two participant groups and stable/volatile contexts are due to different weightings of multiple decision strategies. In contrast, the other three models posit that behavioral differences mainly arise via different learning rates between groups and contexts.

## The MOS model

The key signature of the MOS model is that its policy consists of a mixture of three strategies: EU, MO, and HA. Among many possible variants of the MOS models, this particular three-strategy configuration was chosen as the representative model because it best accounts for human behavioral data (*Figure 3—figure supplement 1*).

The EU strategy postulates that human agents rationally calculate the value of each stimulus and use the softmax rule to select an action. In this case, the value of a stimulus should be its expected utility: $m(s)\psi(s)$. The probability of choosing a stimulus $s$ thus follows a softmax function.

$$\pi_{EU}(s \mid \psi, m) = \frac{\exp(\beta\psi(s)m(s))}{\sum_{s'}\exp(\beta\psi(s')m(s'))} \tag{1}$$

where $\beta$ is the inverse temperature. For simplicity, we rewrite *Equation 1* in the following form:

$$\pi_{EU}(s \mid \psi, m) = \text{softmax}(\beta\psi(s)m(s)) \tag{2}$$

Different from the EU strategy, the MO strategy postulates that observers only focus on feedback magnitude $m(s)$, disregarding feedback probability $\psi(s)$. This is certainly an irrational strategy but more economical in terms of cognitive efforts. Feedback magnitudes are explicitly shown with the stimuli in each trial and readily available for related cognitive computation. But feedback probability, as a latent variable, requires trial-by-trial learning and inference, which is more cognitively demanding. The MO strategy is defined as,

$$\pi_{MO}(s \mid m) = \text{softmax}(\beta m(s)) \tag{3}$$

Unlike EU and MO, the HA strategy depends on neither feedback magnitude $m(s)$ nor feedback probability $\psi(s)$. The HA strategy reflects the tendency to repeat previous frequent choices. This tendency reflects the habit of choosing a stimulus, a phenomenon called perseveration in literature (*Gershman, 2020*; *Wood and Rünger, 2016*). For example, if an agent chooses stimulus 1 more often in past trials, she will form a preference for stimulus 1 in future trials. We constructed it as a Bernoulli distribution over the two stimuli $\pi_{HA}(s)$. The trial-by-trial update rule of $\pi_{HA}(s)$ will be detailed in *Equation 5-6* below.

We implemented the hybrid policy of a linear mixture of the three strategies following the methods used in *Daw et al., 2011*,

$$\pi(s \mid \psi, m, \pi_{HA}) = w_{EU}\pi_{EU}(s \mid \psi, m) + w_{MO}\pi_{MO}(s \mid m) + w_{HA}\pi_{HA}(s) \tag{4}$$

where $w_{EU}$, $w_{MO}$, and $w_{HA}$ are the weighting parameters of each strategy. The three weighting parameters should be summed to 1, i.e., $w_{EU} + w_{MO} + w_{HA} = 1$. We can thus describe the policy an observer adopted just by examining the weighting parameters. Formulating the hybrid model in this way improves the interpretability of the weighting parameters because all three decision strategies are constructed in a Bernoulli format.

Next, we solve the challenge of probabilistic learning. Two distributions — the feedback probability and the habit—are learned and updated in a trial-by-trial fashion. We updated the feedback probability in a Rescorla-Wagner format (*Rescorla, 1972*):

$$\psi(s_1) = \psi(s_1) + \alpha_\psi(O(s_1) - \psi(s_1))$$
$$\psi(s_2) = 1 - \psi(s_1) \tag{5}$$

where $\alpha_\psi$ is the learning rate for feedback probability. $O(\cdot)$ is an indicator function that returns 1 at the true feedback stimulus and 0 otherwise. To keep consistent with *Gagne et al., 2020*, we also explored the valence-specific learning rate,

$$\alpha_\psi = \begin{cases} \alpha_{\psi+}, & \text{for } (O(s_1) - \psi(s_1)) > 0 \\ \alpha_{\psi-}, & \text{for } (O(s_1) - \psi(s_1)) < 0 \end{cases} \tag{6}$$

$\alpha_{\psi+}$ is the learning rate for better-than-expected outcomes, and $\alpha_{\psi-}$ for worse-than-expected outcomes. It is important to note that *Equation 6* was only applied to the reward context, and the definitions of 'better-than-expected' and 'worse-than-expected' should change accordingly in the aversive context, where we defined $\alpha_{\psi+}$ for $(O(s_1) - \psi(s_1)) < 0$ and $\alpha_{\psi-}$ for $(O(s_1) - \psi(s_1)) > 0$.

In a similar manner, the habit component is updated.

$$\pi_{HA}(s_1) = \pi_{HA}(s_1) + \alpha_{HA}(A(s_1) - \pi_{HA}(s_1))$$
$$\pi_{HA}(s_2) = 1 - \pi_{HA}(s_1)$$

(7)

where $\alpha_{HA}$ is the learning rate for the habitual strategy. $A(\cdot)$ is also an indicator function that returns 1 for the stimulus chosen at the current trial. Intuitively, the stimulus chosen more often will result in a higher $\pi_{HA}$ for subsequent trials.

We developed two variants of the MOS model: a context-free and a context-dependent variant. The context-free MOS6 has a total of six free parameters $\xi = \{\beta, \alpha_{HA}, \alpha_\psi, w_{EU}, w_{MO}, w_{HA}\}$. This variant does not include the design of a value-specific learning rate. The context-dependent variant MOS22 has a total of 22 free parameters. Among them $\beta$ and $\alpha_{HA}$ are context-free parameters that were held the same across all contexts. Parameters $\{\alpha_{\psi+}, \alpha_{\psi-}, w_{EU}, w_{MO}, w_{HA}\}$ are context-dependent parameters that should be fitted independently to each context.

We fit the context-dependent parameters to each context following a 2 (reward/aversive) × 2 (stable/volatile) factorial structure (*Figure 1A*). Specifically, the five context-dependent parameters, the positive learning rate parameter $\alpha_{\psi+}$, the negative learning rate parameter $\alpha_{\psi-}$, and three strategies weights $w_{EU}, w_{MO}, w_{HA}$ were fit separately to each context. The remaining two parameters $\{\beta, \alpha_{HA}\}$ were held constant across all four experimental contexts for each participant. Thus, there were 22 free parameters (5 context-dependent parameters × 4 conditions + 2 context-free parameters) of the MOS model in each participant.

## The FLR model

The FLR model refers to Model 11 (i.e. the best-fitting model) in *Gagne et al., 2020*. Here, we describe the FLR model using the same notation system as the published paper, which is slightly different from the notations in the MOS model. The FLR model models the probability of selecting stimulus 1 as follows:

$$\pi(s_1 \mid v, \pi_{HA}) = \frac{1}{1 + \exp(-\beta v - \beta_{HA}[\pi_{HA}(s_1) - \pi_{HA}(s_2)])}$$

(8)

where $\beta$ and $\beta_{HA}$ are the inverse temperature parameters of the value of the stimulus 1 and the HA strategy, respectively. The value of stimulus 1 represents the advantage of $s_1$ over $s_2$,

$$v = \lambda[\psi(s_1) - \psi(s_2)] + (1 - \lambda)\,\text{sign}(m(s_1) - m(s_2))\,|m(s_1) - m(s_2)|^r$$

(9)

where $\lambda$ is the weighting parameter balancing the two terms. The first term $\psi(s_1) - \psi(s_2)$ indicates the feedback probability difference between the two options. The second term, $\text{sign}(m(s_1) - m(s_2))\,|m(s_1) - m(s_2)|^r$, indicates the feedback magnitude differences scaled by a non-linear factor $r$. Intuitively, the value $v$ of $s_1$ can be understood as the weighted sum of the feedback probability differences and the feedback magnitude difference.

During the learning stage, the FLR model learns the feedback probability using the same equations in the MOS model (*Equations 5; 6*). The context-free variant FLR6 has six free parameters $\xi = \{\alpha_{HA}, r, \beta_{HA}, \alpha_\psi, \beta, \lambda\}$. The context-dependent variant FLR22 considers $\{\alpha_{HA}, r\}$ as context-free parameters and $\{\beta_{HA}, \alpha_{\psi+}, \alpha_{\psi-}, \beta, \lambda\}$ as context-dependent parameters, resulting in a total of 22 free parameters.

## The RS model

We adopted the RS model from *Behrens et al., 2007*. The RS model assumes that participants apply the EU strategy but with a subjectively distorted feedback probability $\widetilde{\psi}(s_1)$,

$$\pi\left(s_1 \mid \widetilde{\psi}, m\right) = \frac{1}{1 + \exp\left(-\beta\left[\widetilde{\psi}\left(s_1\right) m\left(s_1\right) - \widetilde{\psi}\left(s_2\right) m\left(s_2\right)\right]\right)} \tag{10}$$

where $\beta$ is the inverse temperature. The distorted probability is calculated by,

$$\begin{aligned}
\widetilde{\psi}\left(s_1\right) &= \max\left[\min\left[\gamma\left(\psi\left(s_1\right) - 0.5\right) + 0.5, 1\right], 0\right] \\
\widetilde{\psi}\left(s_2\right) &= 1 - \widetilde{\psi}\left(s_1\right)
\end{aligned} \tag{11}$$

where the $\gamma$ indicates participants' risk sensitivity. When $\gamma = 1$, a participant has an unbiased risk balance. $\gamma < 1$ and $\gamma > 1$ indicate risk-seeking and risk-aversive tendencies, respectively.

The RS model learns the feedback probability in the same way as the MOS and FLR models (i.e. *Equation 5*). The model did not include the HA strategy. The context-free variant RS3 has a total of three free parameters $\xi = \{\beta, \alpha_\psi, \gamma\}$. The context-dependent variant RS13 considers $\{\beta\}$ as a context-free parameter and $\{\alpha_{\psi+}, \alpha_{\psi-}, \gamma\}$ as context-dependent parameters, resulting in a total of 13 free parameters.

## The PH model

To explicitly incorporate a learning rate adaptation mechanism, we adopt the PH model from *Pearce and Hall, 1980*. This model proposes an adaptive learning rate, as outlined in *Equation 5*.

$$\begin{aligned}
\psi\left(s_1\right) &= \psi\left(s_1\right) + k\alpha_\psi\left(O\left(s_1\right) - \psi\left(s_1\right)\right) \\
\psi\left(s_2\right) &= 1 - \psi\left(s_1\right)
\end{aligned} \tag{12}$$

where $k$ is a scale factor of the learning rate. Each trial the learning rate is updated in accordance with the absolute prediction error,

$$\alpha_\psi = \alpha_\psi + \eta\left(\left|O\left(s_1\right) - \psi\left(s_1\right)\right| - \alpha_\psi\right) \tag{13}$$

where $\eta$ is the step size for the learning rate. We have no knowledge of participants' learning rate values before the experiment, so we need to also fit the initial learning rate value, $\alpha_\psi^0$. The PH model generates a choice through the EU strategy:

$$\pi\left(s_1 \mid \psi, m\right) = \frac{1}{1 + \exp\left(-\beta\left[\psi\left(s_1\right) m\left(s_1\right) - \psi\left(s_2\right) m\left(s_2\right)\right]\right)} \tag{14}$$

The context-free variant PH4 has a total of four free parameters $\xi = \{\alpha_\psi^0, k, \eta, \beta\}$. The context-dependent variant PH17 considers $\{\alpha_\psi^0\}$ as a context-free parameter and as context-dependent parameters, resulting in a total of 17 free parameters.

## Model fitting

Parameters were estimated for each participant via the maximum a posteriori (MAP) method. The objective function to maximize is:

$$\max_\xi \sum_{i=1}^N \log L\left(s_i \mid m_i, O_i, M, \xi\right) + \log p\left(\xi\right) \tag{15}$$

where $\xi$ means the model-free parameters. $M$ is the model and $N$ refers to the number of trials of the participant's behavioral data. $m_i$, $O_i$, and $s_i$ are the presented magnitude, true feedback probability, and participants' responses recorded in each trial.

Parameter estimation was performed using the *Broyden-Fletcher-Goldfarb-Shanno (BFGS)* algorithm in the *scipy.optimize* module in Python. This algorithm provides an approximation of the inverse Hessian matrix for the parameter, a critical component that can be employed in Bayesian model selection (*Rigoux et al., 2014*). In order to use the BFGS algorithm, we reparametrized the model, thereby transforming the original fitting problem into an unconstrained optimization problem. We carefully tuned the parameter priors to ensure that they had little impact on the fitting results. For each participant, we ran the optimization with 40 randomly chosen initial parameters to avoid local minima.

Importantly, to fit the weighting parameters $(w_{EU}, w_{MO}, w_{HA})$ and ensure they summed to 1, we parameterized the weighting parameters as outputs of a softmax function,

$$w_i = \text{softmax}\left(\lambda_i\right) \forall i \in \left\{EU, MO, HA\right\} \tag{16}$$

and fit the logits $\lambda_i$ of the weights. All logits were assumed to be normally distributed with a prior $N(0, 10)$. In the result section, we used both $(w_{EU}, w_{MO}, w_{HA})$ and $(\lambda_{EU}, \lambda_{MO}, \lambda_{HA})$ to represent participants' strategy preferences. Some of the statistical analyses were performed only on $(\lambda_{EU}, \lambda_{MO}, \lambda_{HA})$ because they are normally distributed.

## Simulation details

### Simulate to understand the three strategies

We run simulations to understand the effects of the three strategies on hit rate, hit rate difference, and learning curve. We first used the MOS6 to simulate the learning behaviors of the healthy control group in 100 independent experiments. The parameters were set as $\beta = 10.803$, $\alpha_{HA} = 0.423$, $\alpha_{\psi} = 0.473$, $\lambda_{EU} = 1.138$, $\lambda_{MO} = -1.547$, $\lambda_{HA} = 0.686$, where the first three parameters represent the median across both groups and the latter three weighting parameters are the median across healthy controls. Each simulated experiment consists of two runs, one showing a stable context first and then a volatile context, and vice versa in the other run. This approach results in a total of 200 runs for the healthy control group. The task sequences were randomly generated using the same design *Gagne et al., 2020* used for data collection. Similarly, we repeated all the simulation procedures for the patient group, except that the parameters were set to $\beta = 10.803$, $\alpha_{HA} = 0.423$, $\alpha_{\psi} = 0.473$, $\lambda_{EU} = 0.515$, $\lambda_{MO} = -0.220$, $\lambda_{HA} = 0.094$. Note that we used identical $\{\beta, \alpha_{HA}, \alpha_{\psi}\}$ in both groups and only varied $\{\lambda_{EU}, \lambda_{MO}, \lambda_{HA}\}$ as the median across the patient participants. We used $\pi_{EU}\left(s \mid \psi, m\right)$, $\pi_{MO}\left(s \mid m\right)$, and $\pi_{HA}\left(s\right)$ to evaluate the task performance associated with each strategy (e.g. *Figure 4B–D*). We did not run each strategy completely independently because the HA strategy alone cannot complete the task without learning from decisions previously made by the EU strategy.

### Simulate to explain learning rate adaptation using MOS6

In one simulated experiment, we sampled the four task sequences from the real data. We simulated 20 experiments with the parameters of $\beta = 10.803$, $\alpha_{HA} = 0.423$, $\alpha_{\psi} = 0.473$, $w_{EU} = 0.60$, $w_{MO} = 0.15$, $w_{HA} = 0.25$ to mimic the behavior of the healthy control participants. The first three are the median of the fitted parameters across all participants; the latter three were chosen to approximate the strategy preferences of real healthy control participants (*Figure 4A*). Similarly, we also simulated 20 experiments for the patient group with the identical values of $\beta$, $\alpha_{HA}$, and $\alpha_{\psi}$, but different strategy preferences $w_{EU} = 0.15$, $w_{MO} = 0.60$, $w_{HA} = 0.25$. In other words, the only difference in the parameters of the two groups is the switched $w_{EU}$ and $w_{MO}$. We then fitted the FLR22 to the behavioral data generated by the MOS6 and examined the learning rate differences across groups and volatile contexts (*Figure 6*).

## Acknowledgements

We thank the authors of *Gagne et al., 2020* for sharing their data. This work was supported by the National Key R&D Program of China (2023YFF1204200), the National Natural Science Foundation of China (32100901), the Natural Science Foundation of Shanghai (21ZR1434700), the Research Project of Shanghai Science and Technology Commission (20dz2260300), and the Fundamental Research Funds for the Central Universities (to R.-Y.Z.)

# Additional information

## Funding

| Funder | Grant reference number | Author |
|---|---|---|
| National Key Research and Development Program of China | 2023YFF1204200 | Ru-Yuan Zhang |
| National Natural Science Foundation of China | 32100901 | Ru-Yuan Zhang |
| Natural Science Foundation of Shanghai | 21ZR1434700 | Ru-Yuan Zhang |
| Science and Technology Commission of Shanghai Municipality | 20dz2260300 | Ru-Yuan Zhang |
| Fundamental Research Funds for the Central Universities | | Ru-Yuan Zhang |

The funders had no role in study design, data collection and interpretation, or the decision to submit the work for publication.

## Author contributions

Zeming Fang, Conceptualization, Software, Formal analysis, Validation, Investigation, Visualization, Methodology, Writing - original draft, Writing – review and editing; Meihua Zhao, Ting Xu, Yuhang Li, Hanbo Xie, Peng Quan, Conceptualization, Formal analysis, Methodology, Writing – review and editing; Haiyang Geng, Conceptualization, Formal analysis, Supervision, Project administration, Writing – review and editing; Ru-Yuan Zhang, Conceptualization, Formal analysis, Supervision, Funding acquisition, Validation, Visualization, Methodology, Writing - original draft, Project administration, Writing – review and editing

## Author ORCIDs

Zeming Fang https://orcid.org/0000-0002-8091-4413
Meihua Zhao http://orcid.org/0000-0002-2238-0513
Ting Xu http://orcid.org/0000-0003-1590-9025
Yuhang Li http://orcid.org/0009-0004-5934-2635
Hanbo Xie https://orcid.org/0000-0003-1133-8544
Peng Quan https://orcid.org/0000-0001-5416-536X
Haiyang Geng http://orcid.org/0000-0001-6115-807X
Ru-Yuan Zhang https://orcid.org/0000-0002-0654-715X

## Ethics

Human subjects: Informed consent was obtained for all participants. Procedures for experiment 1 were approved by and complied with the guidelines of the Oxford Central University Research Ethics Committee (protocol numbers: MSD-IDREC-C2-2012-36 and MSD-IDREC-C2-2012-20). Procedures for experiment 2 were approved by and complied with the guidelines of the University of California-Berkeley Committee for the Protection of Human Subjects (protocol ID 2010-12-2638).

Reviewer #2 (Public Review): https://doi.org/10.7554/eLife.93887.3.sa1
Reviewer #3 (Public Review): https://doi.org/10.7554/eLife.93887.3.sa2
Author response https://doi.org/10.7554/eLife.93887.3.sa3

# Additional files

## Supplementary files
• MDAR checklist

## Data availability

All behavioral data are public via Open Science Framework.

The following previously published dataset was used:

| Author(s) | Year | Dataset title | Dataset URL | Database and Identifier |
|---|---|---|---|---|
| Gagne C, Zika O, Dayan P, Bishop SJ | 2020 | Impaired adaptation of learning to contingency volatility in internalizing psychopathology | https://osf.io/8mzuj/ | Open Science Framework, 8mzuj |

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
