## [Editor Report · eLife assessment]

This study provides a novel and **valuable** alternative explanation for volatility-induced changes in choice behavior, commonly attributed to learning-rate adaptations. Through rigorous and comprehensive computational modeling of previously published data, the authors provide **convincing** support for the claim that apparent learning-rate adaptations may instead reflect a mixture of decision strategies. Furthermore, they demonstrate that differential weighting of the optimal decision strategy is predicted by psychopathology common to depression and anxiety. This work should be of interest to a wide range of scientists, including psychologists, neuroscientists, computer scientists, and clinicians.

---

## [Referee Report · Reviewer #2 (Public Review)]

Summary:

Previous research shows that humans tend to adjust learning in environments where stimulus-outcome contingencies become more volatile. This learning rate adaptation is impaired in some psychiatric disorders, such as depression and anxiety. In this tudy the authors reanalyze previously published data on a reversal learning task with two volatility levels. Through a new model they provide some evidence for an alternative explanation whereby the learning rate adaptation is driven by different decision-making strategies and not learning deficits. In particular, they propose that adjusting of learning can be explained by deviations from the optimal decision-making strategy (based on maximizing expected utility) due to response stickiness or focus on reward magnitude. Furthermore, a factor related to general psychopathology of individuals with anxiety and depression negatively correlated with the weight on the optimal strategy and response stickiness, while it correlated positively with the magnitude strategy (a strategy that ignores the probability of outcome).

The main strength of the study is a novel and interesting explanation of an otherwise well-established finding in human reinforcement learning. This proposal is supported by rigorously conducted parameter retrieval and the comparison of the novel model to a wide range of previously published models. The authors explore from many angles, if and why the predictions from the new proposed model are superior to previously applied models.

My previous concerns were addressed in the revised version of the manuscript. I believe that the article now provides a new perspective on a well-established learning effect and offer a novel set of interesting response models that can be applied to a wide array of decision-making problems.

I see two limitations of the study not mentioned in the discussion of the manuscript. First, the task features binary inputs and responses, therefore unexpected uncertainty (volatility) is impossible to differentiate from the uncertainty about outcomes, and exploration is inseparable from random choices. Future work could validate these findings in task designs that allow to distinguish these processes. Second, clinical results are based on a small sample of patients and should be interpreted with this in mind.

---

## [Referee Report · Reviewer #3 (Public Review)]

Summary:

This paper presents a new formulation of a computational model of adaptive learning amid environmental volatility. Using a behavioral paradigm and data set made available by the authors of an earlier publication (Gagne et al., 2020), the new model is found to fit the data well. The model's structure consists of three weighted controllers that influence decisions on the basis of (1) expected utility, (2) potential outcome magnitude, and (3) habit. The model offers an interpretation of psychopathology-related individual differences in decision-making behavior in terms of differences in the relative weighting of the three controllers.

Strengths:

The newly proposed "mixture of strategies" (MOS) model is evaluated relative to the model presented in the original paper by Gagne et al., 2020 (here called the "flexible learning rate" or FLR model) and two other models. Appropriate and sophisticated methods are used for developing, parameterizing, fitting, and assessing the MOS model, and the MOS model performs well on multiple goodness-of-fit indices. Parameters of the model show decent recoverability and offer a novel interpretation for psychopathology-related individual differences. Most remarkably, the model seems to be able to account for apparent differences in behavioral learning rates between high-volatility and low-volatility conditions even with no true condition-dependent change in the parameters of its learning/decision processes. This finding calls into question a class of existing models that attribute behavioral adaptation to adaptive learning rates.

Weaknesses:

The authors have responded to the weaknesses noted previously.

---

## [Author Response]

The following is the authors’ response to the original reviews.

**Public Reviews:**

**Reviewer #1:**
Point 1.1Summary: This paper describes a reanalysis of data collected by Gagne et al. (2020), who investigated how human choice behaviour differs in response to changes in environmental volatility. Several studies to date have demonstrated that individuals appear to increase their learning rate in response to greater volatility and that this adjustment is reduced amongst individuals with anxiety and depression. The present authors challenge this view and instead describe a novel Mixture of Strategies (MOS) model, that attributes individual differences in choice behaviour to different weightings of three distinct decision-making strategies. They demonstrate that the MOS model provides a superior fit to the data and that the previously observed differences between patients and healthy controls may be explained by patients opting for a less cognitively demanding, but suboptimal, strategy.Strengths:The authors compare several models (including the original winning model in Gagne et al., 2020) that could feasibly fit the data. These are clearly described and are evaluated using a range of model diagnostics. The proposed MOS model appears to provide a superior fit across several tests.The MOS model output is easy to interpret and has good face validity. This allows for the generation of clear, testable, hypotheses, and the authors have suggested several lines of potential research based on this.

We appreciate the efforts in understanding our manuscript. This is a good summary.

Point 1.2The authors justify this reanalysis by arguing that learning rate adjustment (which has previously been used to explain choice behaviour on volatility tasks) is likely to be too computationally expensive and therefore unfeasible. It is unclear how to determine how "expensive" learning rate adjustment is, and how this compares to the proposed MOS model (which also includes learning rate parameters), which combines estimates across three distinct decision-making strategies.

We are sorry for this confusion. Actually, our motivation is that previous models only consider the possibility of learning rate adaptation to different levels of environmental volatility. The drawback of previous computational modeling is that they require a large number of parameters in multi-context experiments. We feel that learning rate adaptation may not be the only mechanisms or at least there may exist alternative explanations. Understanding the true mechanisms is particularly important for rehabilitation purposes especially in our case of anxiety and depression. To clarify, we have removed all claims about the learning rate adaptation is “too complex to understand”.

Point 1.3As highlighted by the authors, the model is limited in its explanation of previously observed learning differences based on outcome value. It's currently unclear why there would be a change in learning across positive/negative outcome contexts, based on strategy choice alone.

Thanks for mentioning this limitation. We want to highlight two aspect of work.

First, we developed the MOS6 model primarily to account for the learning rate differences between stable and volatile contexts, and between healthy controls and patients, not for between positive and negative outcomes. In the other words, our model does not eliminate the possibility of different learning rate in positive and negative outcomes.

Second, Figure 3A shows that FLR (containing different learning parameters for positive/negative outcomes) even performed worse than MOS6 (setting identical learning rate for positive/negative outcomes). This result question whether learning rate differences between positive/negative outcomes exist in our dataset.

Action: We now include this limitation in lines 784-793 in discussion:

“The MOS model is developed to offer context-free interpretations for the learning rate differences observed both between stable and volatile contexts and between healthy individuals and patients. However, we also recognize that the MOS account may not justify other learning rate effects based solely on strategy preferences. One such example is the valence-specific learning rate differences, where learning rates for better-than-expected outcomes are higher than those for worse-than-expected outcomes (Gagne et al., 2020). When fitted to the behavioral data, the context-dependent MOS22 model does not reveal valence-specific learning rates (Supplemental Note 4). Moreover, the valence-specific effect was not replicated in the FLR22 model when fitted to the synthesized data of MOS6.”

Point 1.4Overall the methods are clearly presented and easy to follow, but lack clarity regarding some key features of the reversal learning task.Throughout the method the stimuli are referred to as "right" and "left". It's not uncommon in reversal learning tasks for the stimuli to change sides on a trial-by-trial basis or counterbalanced across stable/volatile blocks and participants. It is not stated in the methods whether the shapes were indeed kept on the same side throughout. If this is the case, please state it. If it was not (and the shapes did change sides throughout the task) this may have important implications for the interpretation of the results. In particular, the weighting of the habitual strategy (within the Mixture of Strategies model) could be very noisy, as participants could potentially have been habitual in choosing the same side (i.e., performing the same motor movement), or in choosing the same shape. Does the MOS model account for this?

We are sorry for the confusion. Yes, two shapes indeed changed sides throughout the task. We replaced the “left” and “right” with “stimulus 1” and “stimulus 2”. We also acknowledge the possibility that participants may develop a habitual preference for a particular side, rather than a shape. Due to the counterbalance design, habitual on side will introduce a random selection noise in choices, which should be captured by the MOS model through the inverse temperature parameter.

Point 1.5Line 164: "Participants received points or money in the reward condition and an electric shock in the punishment condition." What determined whether participants received points or money, and did this differ across participants?

Thanks! We have the design clarified in lines 187-188:

“Each participant was instructed to complete two blocks of the volatile reversal learning task, one in the reward context and the other in the aversive context”,

and in lines:

“A total of 79 participants completed tasks in both feedback contexts. Four participants only completed the task in the reward context, while three participants only completed the aversive task.”

Point 1.6Line 167: "The participant received feedback only after choosing the correct stimulus and received nothing else" Is this correct? In Figure 1a it appears the participant receives feedback irrespective of the stimulus they chose, by either being shown the amount 1-99 they are being rewarded/shocked, or 0. Additionally, what does the "correct stimulus" refer to across the two feedback conditions? It seems intuitive that in the reward version, the correct answer would be the rewarding stimulus - in the loss version is the "correct" answer the one where they are not receiving a shock?

Thanks for raising this issue. We removed the term “correct stimulus” and revised the lines 162-166 accordingly:

“Only one of the two stimuli was associated with actual feedback (0 for the other one). The feedback magnitude, ranged between 1-99, is sampled uniformly and independently for each shape from trial to trial. Actual feedback was delivered only if the stimulus associated with feedback was chosen; otherwise, a number “0” was displayed on the screen, signifying that the chosen stimulus returns nothing.”

Point 1.7Line 176: "The whole experiment included two runs each for the two feedback conditions." Does this mean participants completed the stable and volatile blocks twice, for each feedback condition? (i.e., 8 blocks total, 4 per feedback condition).

Thanks! We have removed the term “block”, and now we refer to it as “context”. In particular, we removed phrases like “stable block” and “volatile block” and used “context” instead.

Action: See lines 187-189 for the revised version.

“Each participant was instructed to complete two runs of the volatile reversal learning task, one in the reward context and the other in the aversive context. Each run consisted of 180 trials, with 90 trials in the stable context and 90 in the volatile context (Fig. 1B).”

Point 1.8In the expected utility (EU) strategy of the Mixture or Strategies model, the expected value of the stimulus on each trial is produced by multiplying the magnitude and probability of reward/shock. In Gagne et al.'s original paper, they found that an additive mixture of these components better-captured participant choice behaviour - why did the authors not opt for the same strategy here?

Thanks for asking this. Their strategy basic means the mixture of PF+MO+HA, where PF stands for the feedback probability (e.g., 0.3 or 0.7) without multiplying feedback magnitude. However, ours are EU+MO+HA, where EU stands for feedback probability x feedback magnitude. We did compare these two strategies and the model using their strategy performed much worse than ours (see the red box below).

**Author response image 1. sa3fig1:** Thorough model comparison.

Point 1.9How did the authors account for individuals with poor/inattentive responding, my concern is that the habitual strategy may be capturing participants who did not adhere to the task (or is this impossible to differentiate?).

The current MOS6 model distinguishes between the HA strategy and the inattentive response. Due to the counterbalance design, the HA strategy requires participants to actively track the stimuli on the screen. In contrast, the inattentive responding, like the same motor movement mentioned in Point 1.4, should exhibit random selection in their behavioral data, which should be account by the inverse temperature parameter.

Point 1.10The authors provide a clear rationale for, and description of, each of the computational models used to capture participant choice behaviour.• Did the authors compare different combinations of strategies within the MOS model (e.g., only including one or two strategies at a time, and comparing fit?) I think more explanation is needed as to why the authors opted for those three specific strategies.

We appreciate this great advice. Following your advice, we conducted a thorough model comparisons. Please refer to Figure R1 above. The detailed text descriptions of all the models in Figure R1 are included in Supplemental Note 1.

Point 1.11Please report the mean and variability of each of the strategy weights, per group.

Thanks. We updated the mean of variability of the strategies in lines 490-503:

“We first focused on the fitted parameters of the MOS6 model. We compared the weight parameters (, ,) across groups and conducted statistical tests on their logits (, ,). The patient group showed a ~37% preference towards the EU strategy, which is significantly weaker than the ~50% preference in healthy controls (healthy controls’ : M = 0.991, SD = 1.416; patients’ : M = 0.196, SD = 1.736; t(54.948) = 2.162, p = 0.035, Cohen’s d = 0.509; Fig. 4A). Meanwhile, the patients exhibited a weaker preference (~27%) for the HA strategy compared to healthy controls (~36%) (healthy controls’ : M = 0.657, SD = 1.313; patients’ : M = -0.162, SD = 1.561; t(56.311) = 2.455, p = 0.017, Cohen’s d = 0.574), but a stronger preference for the MO strategy (36% vs. 14%; healthy controls’ : M = -1.647, SD = 1.930; patients’ : M = -0.034, SD = 2.091; t(63.746) = -3.510, p = 0.001, Cohen’s d = 0.801). Most importantly, we also examined the learning rate parameter in the MOS6 but found no group differences (t(68.692) = 0.690, p = 0.493, Cohen’s d = 0.151). These results strongly suggest that the differences in decision strategy preferences can account for the learning behaviors in the two groups without necessitating any differences in learning rate per se.”

Point 1.12The authors compare the strategy weights of patients and controls and conclude that patients favour more simpler strategies (see Line 417), based on the fact that they had higher weights for the MO, and lower on the EU.(1) However, the finding that control participants were more likely to use the habitual strategy was largely ignored. Within the control group, were the participants significantly more likely to opt for the EU strategy, over the HA? (2) Further, on line 467 the authors state "Additionally, there was a significant correlation between symptom severity and the preference for the HA strategy (Pearson's r = -0.285, p = 0.007)." Apologies if I'm mistaken, but does this negative correlation not mean that the greater the symptoms, the less likely they were to use the habitual strategy?I think more nuance is needed in the interpretation of these results, particularly in the discussion.

Thanks. The healthy participants seemed more likely to opt for the EU strategy, although this difference did not reach significance (paired-t(53) = 1.258, p = 0.214, Cohen’s d = 0.242). We systematically explore the role of HA. Compared to the MO, the HA saves cognitive resources but yields a significantly higher hit rate (Fig. 4A). Therefore, a preference for the HA over the MO strategy may reflect a more sophisticated balance between reward and complexity within an agent: when healthier subjects run out of cognitive resources for the EU strategy, they will cleverly resort to the HA strategy, adopting a simpler strategy but still achieving a certain level of hit rate. This explains the negative symptom-HA correlation. As clever as the HA strategy is, it is not surprising that the health control participants opt more for the HA during decision-making.

However, we are cautious to draw strong conclusion on (1) non-significant difference between EU and HA within health controls and (2) the negative symptom-HA correlation. The reason is that the MOS22, the context-dependent variant, (1) exhibited a significant higher preference for EU over HA (paired-t(53) = 4.070, p < 0.001, Cohen’s d = 0.825) and (2) did not replicate this negative correlation (Supplemental Information Figure S3).

Action: Simulation analysis on the effects of HA was introduced in lines 556-595 and Figure 4. We discussed the effects of HA in lines 721-733:

“Although many observed behavioral differences can be explained by a shift in preference from the EU to the MO strategy among patients, we also explore the potential effects of the HA strategy. Compared to the MO, the HA strategy also saves cognitive resources but yields a significantly higher hit rate (Fig. 4A). Therefore, a preference for the HA over the MO strategy may reflect a more sophisticated balance between reward and complexity within an agent (Gershman, 2020): when healthier participants exhaust their cognitive resources for the EU strategy, they may cleverly resort to the HA strategy, adopting a simpler strategy but still achieving a certain level of hit rate. This explains the stronger preference for the HA strategy in the HC group (Fig. 3A) and the negative correlation between HA preferences and symptom severity (Fig. 5). Apart from shedding light on the cognitive impairments of patients, the inclusion of the HA strategy significantly enhances the model’s fit to human behavior (see examples in Daw et al. (2011); Gershman (2020); and also Supplemental Note 1 and Supplemental Figure S3).”

Point 1.13Line 513: "their preference for the slowest decision strategy" - why is the MO considered the slowest strategy? Is it not the least cognitively demanding, and therefore, the quickest?

Sorry for the confusion. In Fig. 5C, we conducted simulations to estimate the learning speed for each strategy. As shown below, the MO strategy exhibits a flat learning curve. Our claim on the learning speed was based solely on simulation outcomes without referring to cognitive demands. Note that our analysis did not aim to compare the cognitive demands of the MO and HA strategies directly.

Action: We explain the learning speed of the three strategies in lines 571-581.

Point 1.14The authors argue that participants chose suboptimal strategies, but do not actually report task performance. How does strategy choice relate to the performance on the task (in terms of number of rewards/shocks)? Did healthy controls actually perform any better than the patient group?

Thanks for the suggestion. The answers are: (1) EU is the most rewarding > the HA > the MO (Fig. 5A), and (2) yes healthy controls did actually perform better than patients in terms of hit rate (Fig. 2).

Action: We included additional sections on above analyses in lines 561-570 and lines 397-401.

Point 1.15The authors speculate that Gagne et al. (2020) did not study the relationship between the decision process and anxiety and depression, because it was too complex to analyse. It's unclear why the FLR model would be too complex to analyse. My understanding is that the focus of Gagne's paper was on learning rate (rather than noise or risk preference) due to this being the main previous finding.

Thanks! Yes, our previous arguments are vague and confusing. We have removed all this kind of arguments.

Point 1.16Minor Comments:• Line 392: Modeling fitting > Model fitting• Line 580 reads "The MO and HA are simpler heuristic strategies that are cognitively demanding."- should this read as less cognitively demanding?• Line 517: health > healthy• Line 816: Desnity > density

Sorry for the typo! They have all been fixed.

**Reviewer #2:**
Point 2.1Summary: Previous research shows that humans tend to adjust learning in environments where stimulus-outcome contingencies become more volatile. This learning rate adaptation is impaired in some psychiatric disorders, such as depression and anxiety. In this study, the authors reanalyze previously published data on a reversal-learning task with two volatility levels. Through a new model, they provide some evidence for an alternative explanation whereby the learning rate adaptation is driven by different decision-making strategies and not learning deficits. In particular, they propose that adjusting learning can be explained by deviations from the optimal decision-making strategy (based on maximizing expected utility) due to response stickiness or focus on reward magnitude. Furthermore, a factor related to the general psychopathology of individuals with anxiety and depression negatively correlated with the weight on the optimal strategy and response stickiness, while it correlated positively with the magnitude strategy (a strategy that ignores the probability of outcome).

Thanks for evaluating our paper. This is a good summary.

Point 2.2My main concern is that the winning model (MOS6) does not have an error term (inverse temperature parameter beta is fixed to 8.804).(1) It is not clear why the beta is not estimated and how were the values presented here chosen. It is reported as being an average value but it is not clear from which parameter estimation. Furthermore, with an average value for participants that would have lower values of inverse temperature (more stochastic behaviour) the model is likely overfitting.(2) In the absence of a noise parameter, the model will have to classify behaviour that is not explained by the optimal strategy (where participants simply did not pay attention or were not motivated) as being due to one of the other two strategies.

We apologize for any confusion caused by our writing. We did set the inverse temperature as a free parameter and quantitatively estimate it during the model fitting and comparison. We also created a table to show the free parameters for each models. In the previous manuscript, we did mention “temperature parameter beta is fixed to 8.804”, but only for the model simulation part, which is conducted to interpret some model behaviors.

We agree with the concern that using the averaged value over the inverse temperature could lead to overfitting to more stochastic behaviors. To mitigate this issue, we now used the median as a more representative value for the population during simulation. Nonetheless, this change does not affect our conclusion (see simulation results in Figures 4&6).

Action: We now use the term “free parameter” to emphasize that the inverse temperature was fitted rather than fixed. We also create a new table “Table 1” in line 458 to show all the free parameters within a model. We also update the simulation details in lines 363-391 for more clarifications.

Point 2.3(3) A model comparison among models with inverse temperature and variable subsets of the three strategies (EU + MO, EU + HA) would be interesting to see. Similarly, comparison of the MOS6 model to other models where the inverse temperature parameter is fixed to 8.804.This is an important limitation because the same simulation as with the MOS model in Figure 3b can be achieved by a more parsimonious (but less interesting) manipulation of the inverse temperature parameter.

Thanks, we added a comparison between the MOS6 and the two lesion models (EU + MO, EU + HA). Please refer to the figure below and Point 1.8.

We also realize that the MO strategy could exhibit averaged learning curves similar to random selection. To confirm that patients' slower learning rates are due to a preference for the MO strategy, we compared the MOS6 model with a variant (see the red box below) in which the MO strategy is replaced by Random (RD) selection that assigns a 0.5 probability to both choices. This comparison showed that the original MOS6 model with the MO strategy better fits human data.

**Author response image 2. sa3fig2:** 

Point 2.4Furthermore, the claim that the EU represents an optimal strategy is a bit overstated. The EU strategy is the only one of the three that assumes participants learn about the stimulus-outcomes contingencies. Higher EU strategy utilisation will include participants that are more optimal (in maximum utility maximisation terms), but also those that just learned better and completely ignored the reward magnitude.

Thank you for your feedback. We have now revised the paper to remove all statement about “EU strategy is the optimal” and replaced by “EU strategy is rewarding but complex”. We agree that both the EU strategy and the strategy only focusing on feedback probability (i.e., ignoring the reward magnitude, refer to as the PF strategy) are rewarding but complex beyond two simple heuristics. We also included the later strategy in our model comparisons (see the next section Point 2.5).

Point 2.5The mixture strategies model is an interesting proposal, but seems to be a very convoluted way to ask: to what degree are decisions of subjects affected by reward, what they've learned, and response stickiness? It seems to me that the same set of questions could be addressed with a simpler model that would define choice decisions through a softmax with a linear combination of the difference in rewards, the difference in probabilities, and a stickiness parameter.

Thanks for suggesting this model. We did include the proposed linear combination models (see “linear comb.” in the red box below) and found that it performed significantly worse than the MOS6.

Action: We justified our model selection criterion in the Supplemental Note 1.

**Author response image 3. sa3fig3:** 

Point 2.6Learning rate adaptation was also shown with tasks where decision-making strategies play a less important role, such as the Predictive Inference task (see for instance Nassar et al, 2010). When discussing the merit of the findings of this study on learning rate adaptation across volatility blocks, this work would be essential to mention.

Thanks for mentioning this great experimental paradigm, which provides an ideal solution for disassociating the probability learning and decision process. We have discussed about this paradigm as well as the associated papers in discussion lines 749-751, 763-765, and 796-801.

Point 2.7Minor mistakes that I've noticed:Equation 6: The learning rate for response stickiness is sometimes defined as alpha_AH or alpha_pi.Supplementary material (SM) Contents are lacking in Note1. SM talks about model MOS18, but it is not defined in the text (I am assuming it is MOS22 that should be talked about here).

Thanks! Fixed.

**Reviewer #3:**
Point 3.1Summary: This paper presents a new formulation of a computational model of adaptive learning amid environmental volatility. Using a behavioral paradigm and data set made available by the authors of an earlier publication (Gagne et al., 2020), the new model is found to fit the data well. The model's structure consists of three weighted controllers that influence decisions on the basis of (1) expected utility, (2) potential outcome magnitude, and (3) habit. The model offers an interpretation of psychopathology-related individual differences in decision-making behavior in terms of differences in the relative weighting of the three controllers.Strengths: The newly proposed "mixture of strategies" (MOS) model is evaluated relative to the model presented in the original paper by Gagne et al., 2020 (here called the "flexible learning rate" or FLR model) and two other models. Appropriate and sophisticated methods are used for developing, parameterizing, fitting, and assessing the MOS model, and the MOS model performs well on multiple goodness-of-fit indices. The parameters of the model show decent recoverability and offer a novel interpretation for psychopathology-related individual differences. Most remarkably, the model seems to be able to account for apparent differences in behavioral learning rates between high-volatility and low-volatility conditions even with no true condition-dependent change in the parameters of its learning/decision processes. This finding calls into question a class of existing models that attribute behavioral adaptation to adaptive learning rates.

Thanks for evaluating our paper. This is a good summary.

Point 3.2(1) Some aspects of the paper, especially in the methods section, lacked clarity or seemed to assume context that had not been presented. I found it necessary to set the paper down and read Gagne et al., 2020 in order to understand it properly.(3) Clarification-related suggestions for the methods section:- Explain earlier that there are 4 contexts (reward/shock crossed with high/low volatility). Lines 252-307 contain a number of references to parameters being fit separately per context, but "context" was previously used only to refer to the two volatility levels.

Action: We have placed the explanation as well as the table about the 4 contexts (stable-reward/stable-aversive/volatile-reward/volatile-aversive) earlier in the section that introduces the experiment paradigm (lines 177-186):

“Participants was supposed to complete this learning and decision-making task in four experimental contexts (Fig. 1A), two feedback contexts (reward or aversive) two volatility contexts (stable or volatile). Participants received points in the reward context and an electric shock in the aversive context. The reward points in the reward context were converted into a monetary bonus by the end of the task, ranging from £0 to £10. In the stable context, the dominant stimulus (i.e., a certain stimulus induces the feedback with a higher probability) provided a feedback with a fixed probability of 0.75, while the other one yielded a feedback with a probability of 0.25. In the volatile context, the dominant stimulus’s feedback probability was 0.8, but the dominant stimulus switched between the two every 20 trials. Hence, this design required participants to actively learn and infer the changing stimulus-feedback contingency in the volatile context.”

- It would be helpful to provide an initial outline of the four models that will be described since the FLR, RS, and PH models were not foreshadowed in the introduction. For the FLR model in particular, it would be helpful to give a narrative overview of the components of the model before presenting the notation.

Action: We now include an overview paragraph in the section of computation model to outline the four models as well as the hypotheses constituted in the model (lines 202-220).

- The subsection on line 343, describing the simulations, lacks context. There are references to three effects being simulated (and to "the remaining two effects") but these are unclear because there's no statement in this section of what the three effects are.- Lines 352-353 give group-specific weighting parameters used for the stimulations of the HC and PAT groups in Figure 4B. A third, non-group-specific set of weighting parameters is given above on lines 348-349. What were those used for?- Line 352 seems to say Figure 4A is plotting a simulation, but the figure caption seems to say it is plotting empirical data.

These paragraphs has been rewritten and the abovementioned issues have been clarified. See lines 363-392.

Point 3.2(2) There is little examination of why the MOS model does so well in terms of model fit indices. What features of the data is it doing a better job of capturing? One thing that makes this puzzling is that the MOS and FLR models seem to have most of the same qualitative components: the FLR model has parameters for additive weighting of magnitude relative to probability (akin to the MOS model's magnitude-only strategy weight) and for an autocorrelative choice kernel (akin to the MOS model's habit strategy weight). So it's not self-evident where the MOS model's advantage is coming from.

An intuitive understanding of the FLR model is that it estimates the stimuli value through a linear combination of probability feedback (PF, [ψ(s1)−ψ(s2)] ) and (non-linear) magnitude (|m(s1)−m(s2)|r) . See equation:v=λ[ψ(s1)−ψ(s2)]+(1−λ)sign⁡(m(s1)−m(s2))|m(s1)−m(s2)|r

Also, the FLR model include the mechanisms of HA as:π(s1∣v,πHA)=11+exp⁡(−βv−βHA[πHA(s1)−πHA(s2)])

In other words, FLR model considers the mechanisms about the probability of feedback (PF)+MO+HA (see Eq. XX in the original study), but our MOS considers the mechanisms of EU+MO+HA. The key qualitative difference lies between FLR and MOS is the usage of the expected utility formula (EU) instead the probability of feedback (PF). The advantage of our MOS model has been fully evidenced by our model comparisons, indicating that human participants multiply probability and magnitude rather than only considering probability. The EU strategy has also been suggested by a large pile of literature (Gershman et al., 2015; Von Neumann & Morgenstern, 1947).

Making decisions based on the multiplication of feedback probability and magnitude can often yield very different results compared to decisions based on a linear combination of the two, especially when the two magnitudes have a small absolute difference but a large ratio. Let’s consider two cases:

(1) Stimulus 1: ψ1=0.75,m1=0.1 vs. Stimulus 2: ψ1=0.25,m1=0.4

(2) Stimulus 1: ψ1=0.75,m1=0.2 vs. Stimulus 2:ψ1=0.25,m1=0.8

The EU strategy may opt for stimulus 2 in both cases, since stimulus 2 always has a larger expected value. However, it is very likely for the PF+MO to choose stimulus 1 in the first case. For example, when λ=0.5,λψ1+(1−λ)m1=0.425>λψ2+(1−λ)m2=0.325 . If we want the PF+MO to also choose stimulus to align with the EU strategy, we need to increase the weight on magnitude (1−λ) . Note that in this example we divided the magnitude value by 100 to ensure that probability and magnitude are on the same scale to help illustration.

In the dataset reported by Gagne, 2020, the described scenario seems to occur more often in the aversive context than in the reward context. To accurately capture human behaviors, FLR22 model requires a significantly larger weight for magnitude in the aversive context (1−λ≈0.4) than in the reward context (1−λ≈0.2) . Interestingly, when the weights for magnitude in different contexts are forced to be equal, the model (FLR6) fails, exhibiting an almost chance-level performance throughout learning (Fig. 3E, G). In contrast, the MOS6 model, and even the RS3 model, exhibit good performance using one identical set of parameters across contexts. Both MOS6 and RS3 include the EU strategy during decision-making. These findings suggest humans make decisions using the EU strategy rather than PF+MO.

The focus of our paper is to present that a good-enough model can interpret the same dataset in a completely different perspective, not necessarily to explore improvements for the FLR model.

Point 3.3One of the paper's potentially most noteworthy findings (Figure 5) is that when the FLR model is fit to synthetic data generated by the expected utility (EU) controller with a fixed learning rate, it recovers a spurious difference in learning rate between the volatile and stable environments. Although this is potentially a significant finding, its interpretation seems uncertain for several reasons:- According to the relevant methods text, the result is based on a simulation of only 5 task blocks for each strategy. It would be better to repeat the simulation and recovery multiple times so that a confidence interval or error bar can be estimated and added to the figure.- It makes sense that learning rates recovered for the magnitude-oriented (MO) strategy are near zero, since behavior simulated by that strategy would have no reason to show any evidence of learning. But this makes it perplexing why the MO learning rate in the volatile condition is slightly positive and slightly greater than in the stable condition.- The pure-EU and pure-MO strategies are interpreted as being analogous to the healthy control group and the patient group, respectively. However, the actual difference in estimated EU/MO weighting between the two participant groups was much more moderate. It's unclear whether the same result would be obtained for a more empirically plausible difference in EU/MO weighting.- The fits of the FLR model to the simulated data "controlled all parameters except for the learning rate parameters across the two strategies" (line 522). If this means that no parameters except learning rate were allowed to differ between the fits to the pure-EU and pure-MO synthetic data sets, the models would have been prevented from fitting the difference in terms of the relative weighting of probability and magnitude, which better corresponds to the true difference between the two strategies. This could have interfered with the estimation of other parameters, such as learning rate.- If, after addressing all of the above, the FLR model really does recover a spurious difference in learning rate between stable and volatile blocks, it would be worth more examination of why this is happening. For example, is it because there are more opportunities to observe learning in those blocks?I would recommend performing a version of the Figure 5 simulations using two sets of MOS-model parameters that are identical except that they use healthy-control-like and patient-like values of the EU and MO weights (similar to the parameters described on lines 346-353, though perhaps with the habit controller weight equated). Then fit the simulated data with the FLR model, with learning rate and other parameters free to differ between groups. The result would be informative as to (1) whether the FLR model still misidentifies between-group strategy differences as learning rate differences, and (2) whether the FLR model still identifies spurious learning rate differences between stable and volatile conditions in the control-like group, which become attenuated in the patient-like group.

Many thanks for this great advice. Following your suggestions, we now conduct simulations using the median of the fitted parameters. The representations for healthy controls and patients have identical parameters, except for the three preference parameters; moreover, the habit weights are not controlled to be equal. 20 simulations for each representative, each comprising 4 task sequences sampled from the behavioral data. In this case, we could create error bars and perform statistical tests. We found that the differences in learning rates between stable and volatile conditions, as well as the learning rate adaptation differences between healthy controls and patients, still persisted.

Combined with the discussion in Point 3.2, we justify why a mixture-of-strategy can account for learning rate adaptation as follow. Due to (unknown) differences in task sequences, the MOS6 model exhibits more MO-like behaviors due to the usage of the EU strategy. To capture this behavior pattern, the FLR22 model has to increase its weighting parameter 1-λ for magnitude, which could ultimately drive the FLR22 to adjust the fitted learning rate parameters, exhibiting a learning rate adaptation effect. Our simulations suggest that estimating learning rate just by model fitting may not be the only way to interpret the data.

Action: We included the simulation details in the method section (lines 381-lines 391)

“In one simulated experiment, we sampled the four task sequences from the real data. We simulated 20 experiments with the parameters of β=10.803,αHA=0.423,αΨ=0.473,wEU=0.60,wMO=0.15,wHA=0.25 to mimic the behavior of the healthy control participants. The first three are the median of the fitted parameters across all participants; the latter three were chosen to approximate the strategy preferences of real health control participants (Figure 4A). Similarly, we also simulated 20 experiments for the patient group with the identical values of β,αHA , and αΨ , but different strategy preferences wEU=0.15,wMO=0.60,wHA=0.25 . In other words, the only difference in the parameters of the two groups is the switched wEU and wMO . We then fitted the FLR22 to the behavioral data generated by the MOS6 and examined the learning rate differences across groups and volatile contexts (Fig. 6). ”

Point 3.4Figure 4C shows that the habit-only strategy is able to learn and adapt to changing contingencies, and some of the interpretive discussion emphasizes this. (For instance, line 651 says the habit strategy brings more rewards than the MO strategy.) However, the habit strategy doesn't seem to have any mechanism for learning from outcome feedback. It seems unlikely it would perform better than chance if it were the sole driver of behavior. Is it succeeding in this example because it is learning from previous decisions made by the EU strategy, or perhaps from decisions in the empirical data?

Yes, the intuition is that the HA strategy seems to show no learning mechanism. But in reality, it yields a higher hit rate than MO by simply learning from previous decisions made by the EU strategy. We run simulations to confirm this (Figure 4B).

Point 3.5For the model recovery analysis (line 567), the stated purpose is to rule out the possibility that the MOS model always wins (line 552), but the only result presented is one in which the MOS model wins. To assess whether the MOS and FLR models can be differentiated, it seems necessary also to show model recovery results for synthetic data generated by the FLR model.

Sure, we conducted a model recovery analysis that include all models, and it demonstrates that MOS and FLR can be fully differentiated. The results of the new model recovery analysis were shown in Fig. 7.

Point 3.6To the best of my understanding, the MOS model seems to implement valence-specific learning rates in a qualitatively different way from how they were implemented in Gagne et al., 2020, and other previous literature. Line 246 says there were separate learning rates for upward and downward updates to the outcome probability. That's different from using two learning rates for "better"- and "worse"-than-expected outcomes, which will depend on both the direction of the update and the valence of the outcome (reward or shock). Might this relate to why no evidence for valence-specific learning rates was found even though the original authors found such evidence in the same data set?

Thanks. Following the suggestion, we have corrected our implementation of valence-specific learning rate in all models (see lines 261-268).

“To keep consistent with Gagne et al., (2020), we also explored the valence-specific learning rate,αψ={αψ+, for (O(s1)−ψ(s1))>0αψ−, for (O(s1)−ψ(s1))<0

αΨ+ is the learning rate for better-than-expected outcome, and αΨ− for worse-than-expected outcome. It is important to note that Eq. 6 was only applied to the reward context, and the definitions of “better-than-expected” and “worse-than-expected” should change accordingly in the aversive context, where we defined αψ+ for (o(s1)−Ψ(s1)) and αΨ− for (o(s1)−ψ(s1))>0 .

No main effect of valence on learning rate was found (see Supplemental Information Note 3)

Point 3.7The discussion (line 649) foregrounds the finding of greater "magnitude-only" weights with greater "general factor" psychopathology scores, concluding it reflects a shift toward simplifying heuristics. However, the picture might not be so straightforward because "habit" weights, which also reflect a simplifying heuristic, correlated negatively with the psychopathology scores.

Thanks. In contrast the detrimental effects of “MO”, “habit” is actually beneficial for the task. Please refer to Point 1.12.

Point 3.8The discussion section contains some pejorative-sounding comments about Gagne et al. 2020 that lack clear justification. Line 611 says that the study "did not attempt to connect the decision process to anxiety and depression traits." Given that linking model-derived learning rate estimates to psychopathology scores was a major topic of the study, this broad statement seems incorrect. If the intent is to describe a more specific step that was not undertaken in that paper, please clarify. Likewise, I don't understand the justification for the statement on line 615 that the model from that paper "is not understandable" - please use more precise and neutral language to describe the model's perceived shortcomings.

Sorry for the confusion. We have removed all abovementioned pejorative-sounding comments.

Point 3.94. Minor suggestions:- Line 114 says people with psychiatric illness "are known to have shrunk cognitive resources" - this phrasing comes across as somewhat loaded.

Thanks. We have removed this argument.

- Line 225, I don't think the reference to "hot hand bias" is correct. I understand hot hand bias to mean overestimating the probability of success after past successes. That's not the same thing as habitual repetition of previous responses, which is what's being discussed here.

Response: Thanks for mentioning this. We have removed all discussions about “hot hand bias”.

- There may be some notational inconsistency if alpha_pi on line 248 and alpha_HA on line 253 are referring to the same thing.

Thanks! Fixed!

- Check the notation on line 285 - there may be some interchanging of decimals and commas.

Thanks! Fixed!

Also, would the interpretation in terms of risk seeking and risk aversion be different for rewarding versus aversive outcomes?

Thanks for asking. If we understand it correctly, risk seeking and risk aversion mechanisms are only present in the RS models, which show clearly worse fitting performance. We thus decide not to overly interpret the fitted parameters in the RS models.

- Line 501, "HA and PAT groups" looks like a typo.- In Figure 5, better graphical labeling of the panels and axes would be helpful.

Response: Thanks! Fixed!

REFERENCES

Daw, N. D., Gershman, S. J., Seymour, B., Dayan, P., & Dolan, R. J. (2011). Model-based influences on humans' choices and striatal prediction errors. Neuron, 69(6), 1204-1215.

Gagne, C., Zika, O., Dayan, P., & Bishop, S. J. (2020). Impaired adaptation of learning to contingency volatility in internalizing psychopathology. Elife, 9.

Gershman, S. J. (2020). Origin of perseveration in the trade-off between reward and complexity. Cognition, 204, 104394.

Gershman, S. J., Horvitz, E. J., & Tenenbaum, J. B. (2015). Computational rationality: A converging paradigm for intelligence in brains, minds, and machines. Science, 349(6245), 273-278.

Von Neumann, J., & Morgenstern, O. (1947). Theory of games and economic behavior, 2nd rev.